# Single-cell atlas of early chick development reveals gradual segregation of neural crest lineage from the neural plate border during neurulation

**Ruth M Williams[1,2], Martyna Lukoseviciute[2†], Tatjana Sauka-Spengler[2], Marianne E Bronner[1]\***

[1]California Institute of Technology, Division of Biology and Biological engineering, Pasadena, United States; [2]University of Oxford, MRC Weatherall Institute of Molecular Medicine, Radcliffe Department of Medicine, Oxford, United Kingdom

**Abstract** The epiblast of vertebrate embryos is comprised of neural and non-neural ectoderm, with the border territory at their intersection harboring neural crest and cranial placode progenitors. Here, we a generate single-cell atlas of the developing chick epiblast from late gastrulation through early neurulation stages to define transcriptional changes in the emerging 'neural plate border' as well as other regions of the epiblast. Focusing on the border territory, the results reveal gradual establishment of heterogeneous neural plate border signatures, including novel genes that we validate by fluorescent in situ hybridization. Developmental trajectory analysis infers that segregation of neural plate border lineages only commences at early neurulation, rather than at gastrulation as previously predicted. We find that cells expressing the prospective neural crest marker *Pax7* contribute to multiple lineages, and a subset of premigratory neural crest cells shares a transcriptional signature with their border precursors. Together, our results suggest that cells at the neural plate border remain heterogeneous until early neurulation, at which time progenitors become progressively allocated toward defined neural crest and placode lineages. The data also can be mined to reveal changes throughout the developing epiblast.

**\*For correspondence:**
mbronner@caltech.edu

**Present address:** [†]Department of Cell and Molecular Biology, Karolinska Institutet, Stockholm, Sweden

## Introduction

During gastrulation, the ectoderm layer of the chordate embryo becomes segregated into the neural plate and the surrounding non-neural ectoderm. The neural plate ultimately generates the central nervous system (CNS), whereas the surrounding non-neural ectoderm forms the epidermis of the skin as well as the epithelial lining of the mouth and nasal cavities. At the interface of these tissues is a territory referred to as the 'neural plate border' which in vertebrates contains precursors of neural crest, neural, and placodal lineages (*Ezin et al., 2009*; *Streit, 2002*). Neural crest and ectodermal placodes share numerous common features including the ability to migrate or invaginate and form multiple cell types. During neurulation, neural crest cells come to reside within the dorsal neural tube where they undergo an epithelial-to-mesenchymal transition (EMT). Subsequently, they delaminate and migrate throughout the embryo, settle at their final destinations and differentiate into numerous derivatives including neurons and glia of the peripheral nervous system as well as cartilage, bone, and connective tissues elements of the head and face. Like neural crest cells, cranial placode cells become internalized, and then differentiate into sensory neurons and sense organs (nose, ears, lens) of the head. Aberrant neural crest or placode development causes a number of developmental disorders affecting craniofacial structures (*Siismets and Hatch, 2020*;

*Vega-Lopez et al., 2018*), the enteric nervous system (e.g. Hirschsprung's disease) (*Butler Tjaden and Trainor, 2013*), and the heart (e.g. Persistent Truncus Arteriosus; CHARGE syndrome) (*Pauli et al., 2017*; *Gandhi et al., 2020*). Furthermore, a number of malignancies, including melanoma, neuroblastoma. and glioma, are known to arise from neural crest derivatives (*Tomolonis et al., 2018*).

An ongoing question is whether individual neural plate border cells are specified toward a particular lineage (i.e. neural crest, placode, or CNS) or if they have the potential to become any cell type that arises from the border. It has been suggested that progenitors of these different lineages may be regionalized within the neural plate border, with the more lateral cells contributing to the placodes and the more medial region giving rise to neural and neural crest cells (*Schlosser, 2008*). Such segregation has been proposed to result from the influence of graded expression of signalling factors emanating from surrounding tissues, for example Wnts and FGFs, on multipotent neural plate border cells (*Schille and Schambony, 2017*). Alternatively, individual neural plate border cells predetermined toward a particular lineage may be intermingled within the border. A recent study in the chick (*Roellig et al., 2017*) showed that while medial and lateral regions of the border can be discerned by lineage markers, there is significant co-expression of markers characteristic of multiple lineages (neural crest, neural plate, and placodal) across the epiblast. Moreover, this overlap of lineage markers within the neural plate border is maintained from gastrulation through neurulation suggesting that these cells may maintain plasticity through neurulation stages. Thus, while some neural plate border cells may be predisposed toward a particular fate, others retain the ability to generate multiple lineages. However, the timing at which neural plate border cells emerge and become distinguishable from neural and non-neural ectoderm has not been conclusively characterized, complicating the assessment of cell heterogeneity within the neural plate border territory. Here, we provide single-cell transcriptomes of the developing chick epiblast from late gastrulation through early neurulation, allowing us to annotate the neural plate border region and explore heterogeneity therein. The results expand knowledge of transcriptional changes in the border as a function of time, revealing the full complexity of co-expressed genes in this multipotent tissue. Moreover, by using whole epiblast tissue, we have generated the first single-cell atlas of early chicken development enabling a contextual view of neural plate border emergence from surrounding tissues.

One study using an ex ovo culturing method of explants from the chicken embryos proposed that a pre-neural plate border region is established as early as stage Hamburger and Hamilton (HH) 3 (*Prasad et al., 2020*). In addition, an in vitro model to derive neural crest cells from human embryonic stem cells shows that pre-border genes can be induced by Wnt signaling (*Leung et al., 2016*). However, these observations have not been thoroughly addressed in vivo. Therefore, the questions of how and when neural plate border cells establish/retain multipotency and the comprehensive gene dynamics underlying these processes remain open.

Single-cell RNA-sequencing (scRNA-seq) provides a unique platform to address the intriguing question of when a neural plate border transcriptional signature arises in an in vivo context. To this end, we examined single-cell transcriptomes from the epiblast of gastrulating to neurulating chick embryos to determine the time course of emergence of the neural plate border. The chick represents an ideal system for these studies since avian embryos develop as a flat blastodisc at the selected time points, highly reminiscent of human development at comparable stages. Interestingly, our results show that the border is not transcriptionally distinct until the beginning of neurulation (HH7), when neural plate border markers define a discrete subcluster of the ectoderm. Furthermore, RNA velocity measurements imply that segregation of the neural plate border commences at HH6, but is more profoundly underway at HH7, with definitive neural crest clusters only emerging in the elevating neural folds. Velocity analysis also suggests that *Pax7+* cells are not restricted to a neural crest fate but rather are capable of giving rise to all derivatives of the neural plate border. The data also reveal numerous novel factors dynamically expressed across the developing epiblast as well as indicating putative drivers of neural plate border trajectories. Taken together, the data reveal dynamic changes in an emerging neural plate border that becomes progressively segregated into defined lineages, that is complete only late in neurulation. Furthermore, the data can be further interrogated to examine transcriptional changes in other regions emerging from the epiblast.

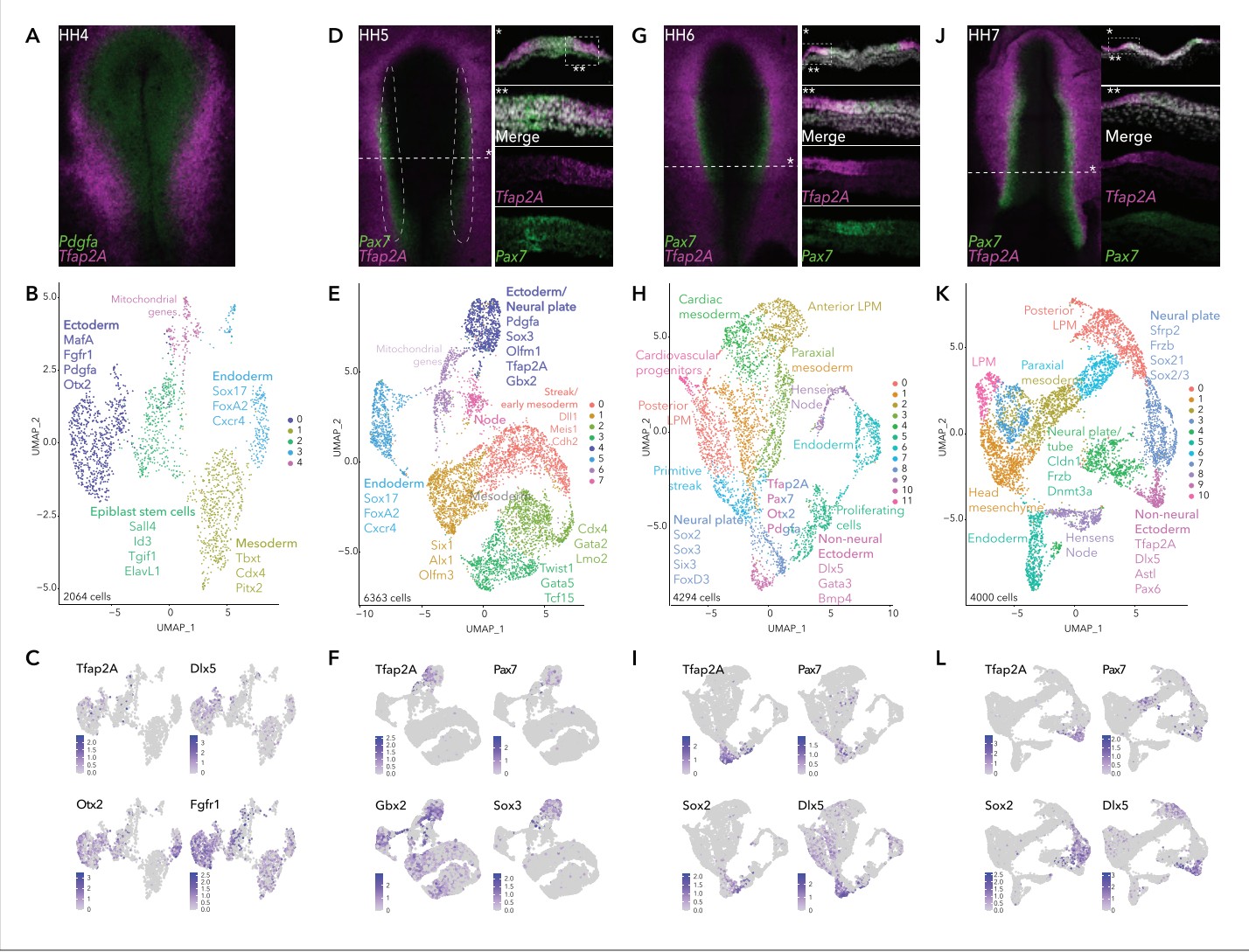

**Figure 1.** Single-cell RNA-seq of avian epiblast from HH4 through HH7. (**A**) *Tfap2A* expression at HH4, detected by HCR. (**B**) UMAP plot depicting five clusters resolved from 2064 epiblast cells at HH4. (**C**) Feature plots of selected genes in HH4 clusters. (**D**) *Pax7* and *Tfap2A* expression at HH5 detected by HCR. (**E**) UMAP plot depicting eight clusters resolved from 6363 cells from dissected neural plate border regions at HH5, black dotted region in (D). (**F**) Feature plots of selected genes in HH5 clusters. (**G**) *Pax7* and *Tfap2A* expression at HH6 detected by HCR. (**H**) UMAP plot depicting 12 clusters resolved from 4294 epiblast cells at HH6. (**I**) Feature plots of selected genes in HH6 clusters. (**J**) *Pax7* and *Tfap2A* expression at HH7 detected by HCR. (**K**) UMAP plot depicting 11 clusters resolved from 4000 epiblast cells at HH7. (**L**) Feature plots of selected genes in HH7 clusters.

The online version of this article includes the following figure supplement(s) for figure 1:

**Figure supplement 1.** Quality control and supporting data for HH4 and HH5 10 X data.

**Figure supplement 2.** Quality control and supporting data for HH6 and HH7 10 X data.

## Results

### Single-cell analysis of the avian epiblast during gastrulation (HH4 – 5)

To resolve the transcriptional signatures of individual neural plate border cells in the context of the developing embryo, we first performed single-cell RNA-seq analysis of the epiblast of gastrulating chick embryos at stages HH4-5 (*Hamburger and Hamilton, 1951*). We used the Chromium 10X platform in order to recover a large number of cells, thus profiling the majority of neural plate border cells. As a reference for emergence of the neural plate border, we used *Pax7* and *Tfap2A* transcription factors, both of which are well-established early markers of the neural plate border (*Basch et al., 2006*; *de Crozé et al., 2011*). *Tfap2A* demarcates the lateral aspect of the neural plate border from

HH4 and is also expressed in the non-neural ectoderm (*Figure 1A*). *Pax7* is progressively enriched in the medial border region from HH5 (*Figure 1D*).

At HH4 (*Figure 1A*), we recovered 2398 cells from eight embryos. Five distinct clusters (2064 cells) were resolved after quality control processing (*Figure 1B*, *Figure 1—figure supplement 1*) and annotated by marker genes identified by single-cell differential expression (SCDE) analysis (*Figure 1—figure supplement 1*). Focusing on the future neural plate border, two ectoderm clusters were recovered (HH4-Cl0, HH4-Cl2), identified by the enrichment in *Cldn1* expression (*Figure 1—figure supplement 1*). HH4-Cl0 was enriched for neural plate markers (*Otx2 and Fgfr1*), and featured low levels of *Tfap2A* and *Dlx5* (*Figure 1C*). Signaling molecules including *Sfrp2*, and *Pdgfa* were also enriched in HH4-Cl0. *Sfrp2* and *Fgfr1* are expressed across the neural plate, extending to the neural plate border (*Chapman et al., 2004*; *Lunn et al., 2007*) and *Pdgfa* was expressed in the posterior epiblast (*Yang et al., 2008*). bHLH transcription factor *MafA*, more commonly associated with pancreatic beta-cell differentiation (*Hang and Stein, 2011*), but also reported in the developing chick neural plate (*Lecoin et al., 2004*), was another marker enriched in HH4-Cl0 (*Figure 1—figure supplement 1*). HH4-Cl2 was less distinctive but enriched in genes associated with pluripotency, such as *Sall4, Tgif1, ElavL1* (*Lee et al., 2015*; *Ye and Blelloch, 2014*; *Zhang et al., 2006*), (*Figure 1—figure supplement 1*) suggesting these may represent residual epiblast stem cells. Other clusters at HH4 were readily identified as mesoderm (HH4-Cl1) or endoderm (HH4-Cl3) as characterized by the expression of *Cdx4* and *Pitx2* for mesoderm and *Sox17*, *FoxA2,* and *Cxcr4* for endoderm (*Figure 1B*, *Figure 1—figure supplement 1*). HH4-Cl2 was enriched for genes associated with pluripotency including *Sall4, Tgif1, ElavL1* (*Lee et al., 2015*; *Ye and Blelloch, 2014*; *Zhang et al., 2006*; *Figure 1—figure supplement 1*). Mitochondrial genes and cell migration factors (*Cxcl12, Itgb1, Tgfbr1*) were enriched in HH4-Cl4 (*Figure 1B*, *Figure 1—figure supplement 1*).

To refine our analysis of the prospective neural plate border, we next performed 10X single-cell RNA-seq on dissected neural plate border regions from HH5 embryos (eight dissections) (*Figure 1D*) yielding 6363 cells and six clusters (*Figure 1E*, *Figure 1—figure supplement 1*). Prospective neural plate border cells were confined to cluster HH5-Cl4, enriched for *Pax7* and *Tfap2A* expression, but also featuring a neural marker *Sox3*, and transcription factor *Gbx2*, which has a known role in neural crest induction in *Xenopus* (*Li et al., 2009*; *Figure 1F*). While neural plate border markers were enriched in HH5-Cl4 cells, co-expression of neural and non-neural ectoderm factors here (*Figure 1—figure supplement 1*) suggests this intermediate region is not yet distinguishable as a unique entity.

## Single-cell analysis of the avian epiblast during neurulation (HH6 - 7)

During the process of neurulation in amniote embryos, the neural plate gradually folds inwards on itself, the border edges elevate forming the neural folds which, as neurulation progresses, fuse at the dorsal midline to form the neural tube. In the chicken embryo, this process is completed by HH8. Therefore, we performed single-cell analysis at HH6 and HH7 to capture changes in the neural plate border as a function of time. While expression of *Pax7* was detectable but low during gastrulation stages (HH4/5), its expression is strongly enhanced by HH6/7 (*Figure 1G/J*), when this transcription factor marks more medial neural plate border cells.

Single-cell analysis at HH6 showed increased complexity revealing 12 distinct clusters from 4294 cells, 12 embryos (*Figure 1H*, *Figure 1—figure supplement 2*). A neural plate cluster (HH6-Cl8) was characterized by neural genes including *Sox2/3*, *Six3*, *Hes5*, and *Otx2* (*Figure 1I*, *Figure 1—figure supplement 2*) as well as neural crest regulators *FoxD3, Ednrb, PdgfA,* and *Gbx2* (*Figure 1—figure supplement 2*), whereas *Tfap2A, Dlx5, and Gata3* were enriched in the cluster HH6-Cl10. *Pax7* and other neural plate border genes *Bmp4* and *Msx1* were heterogeneously detected at the interface of HH6-Cl8 and HH6-Cl10 (*Figure 1I*, *Figure 1—figure supplement 2*), indicating both these clusters harbor neural plate border cells. At HH6 mesodermal cells were found in several clusters: anterior lateral plate mesoderm (HH6-Cl2; *Pitx2, Alx1 OlfmL3, Six1, Twist1*); posterior lateral neural plate (HH6-Cl0; *Gata2, HoxB5, Cdx4*). Cardiac mesoderm markers were enriched in HH6-Cl4 (*Tcf21* and *Gata5*) and HH6-Cl11 (*Lmo2, Ets1, Kdr*). HH6-Cl1 and HH6-Cl3 represented paraxial mesoderm (*Msgn1, Mesp1, Meox1*). Endoderm cells formed HH6-Cl6 (*Sox17, FoxA2*). Hensen's node and primitive streak markers (*Dll1, Fgf8, Noto, Chrd*) were identified in HH6-Cl7 and HH6-Cl9, respectively. We also detected a cluster of cells (HH6-Cl5) with high levels of mitochondrial genes (*Cox1/3*), this cluster

was also enriched for factors associated with highly proliferative cells (*Pdia3, Igf1r*) (*Figure 1—figure supplement 2*).

At HH7 (*Figure 1J*), we identified 11 clusters from 4000 cells, 8 embryos (*Figure 1K, Figure 1—figure supplement 2*). Cells of the neural plate appeared in HH7-Cl3 and were distinct from the non-neural ectoderm cells found in HH7-Cl9 (*Figure 1K/L*). At the interface of these clusters the neural plate border was emerging as depicted by an overlap of *Pax7, Tfap2A, Dlx5, Bmp4, and Msx1* (*Figure 1L* and *Figure 1—figure supplement 2*). Expression of early neural crest genes (*Draxin, Tfap2B*) also emerged in these cells (*Figure 1—figure supplement 2*). In addition to neural markers detected at earlier stages (*Otx2, Sox2, Sox3*), other neural/neural crest genes featured in HH7-Cl3, including *Zeb2* and *Zic2* (*Figure 1—figure supplement 2*). Additional factors, such as *Pax6*, a crucial regulator of eye development (*Liu et al., 2006*) featured in the cluster HH7-Cl9 (*Figure 1—figure supplement 2*). *Zfhx4* (*Figure 1—figure supplement 2*), a zinc finger transcription factor previously observed at later stages in the neural crest and neural tube (*Williams et al., 2019*) featured across HH7-Cl3 and HH7-Cl9 clusters. Aspects of the mesoderm were discernible across several clusters (HH7-Cl0, 1, 2, 6, 7, 10). Including the posterior mesoderm (HH7-Cl0), likely containing neuromesodermal precursors (NMPs) as suggested by the enrichment in *Cdx2/4, Tbxt*, and *Sox2* expression (*Figure 1—figure supplement 2*). HH7-Cl5 was defined by endoderm markers and HH7-Cl8 represented Hensen's node and the primitive streak (*Figure 1—figure supplement 2*).

Overall, across the stages analyzed, we observed a progressive refinement of transcriptional signatures in individual ectoderm clusters reflecting neural versus non-neural lineages. While we identified markers of the neural plate border within multiple clusters, the border itself, surprisingly, is not distinguishable as a unique entity. Highlighting the heterogeneity of cells within the neural plate border, defined by combinatorial gene signatures shared with the surrounding neural plate and non-neural ectoderm.

## Subclustering reveals progressive transcriptional segregation of ectoderm cells

As the chick embryo undergoes gastrulation, ectodermal cells that will become neural, neural plate border, placode, or epidermis remain in the upper layer of the embryo (epiblast), while mesoendodermal cells ingress and internalize at the primitive streak and Hensen's node. To further resolve the transcriptional complexity of the developing neural plate border, we extracted and subclustered the ectodermal clusters at stages HH5, HH6, and HH7. These are designated HH5-Cl4, HH6-Cl8/10, and HH7-Cl3/9 (*Figure 1E, H and K*).

HH5-Cl4 resolved into three closely related clusters (*Figure 2A*). Neural plate (*Sox2, Otx2, Sox21, Six3*) subcluster (HH5-sub-1) versus the non-neural ectoderm (*Tfap2A, Dlx5*) subcluster (HH5-sub-2) (*Figure 2B, Figure 2—figure supplement 1*) were clearly delineated. The third subcluster, HH5-sub-0, contained cells expressing caudal neural plate (*Msx1, Pdgfa, Wnt8A, Pou5f3*) and caudal epiblast genes (*Tbxt, Cdx2/4, Meis1/2*; *Figure 2—figure supplement 1*). *Pax7+* cells were most prominent in HH5-sub-0 (caudal) but a small portion of HH5-sub-2 (non-neural ectoderm) cells also expressed *Pax7* (*Figure 2B*). HH5-sub-0 cells co-expressed other neural plate border genes *Tfap2A, Dlx5, Msx1* and *Bmp4* with *Pax7*, these genes were also present in other clusters (*Figure 2B*).

Similar signatures were found at HH6 (*Figure 2C/D*) but with increasing transcriptional segregation. Here, the neural plate was represented by HH6-sub-1 (*Sox2, Otx2, Six3*; *Figure 2D, Figure 2—figure supplement 1*). Non-neural ectoderm markers (*Tfap2A, Dlx5*) were enriched in HH6-sub-0 (*Figure 2D*). *Pax7+* cells clustered within the HH6-sub-2 (*Figure 2D*), together with caudal epiblast markers (*Tbxt, Cdx2/4*) (*Figure 2—figure supplement 1*); however, *Pax7* expression also featured in HH6-sub-0 and HH6-sub-1, with *Pax7+* cells positioned at the interface of the three subclusters (*Figure 2D*). These *Pax7+* cells at the clusters' interface also expressed *Tfap2A, Msx1, Dlx5*, and *Bmp4* which were enriched in other clusters, as observed at HH5. This demonstrates that the neural plate border cluster is not yet distinct and indicates that neural plate border cells share signatures with other tissues at this stage (*Figure 2D*).

Further transcriptional refinements occurred between the HH5 and HH6 ectoderm subclusters. *Irf6* which was broadly present across HH5 subclusters, was now enriched in HH6-sub-0 (non-neural ectoderm), along with *Grhl3* which was present in a small portion of HH5-sub-2 (non-neural ectoderm; *Figure 2—figure supplement 1*). *FoxD3* expression was barely detectable in cells of HH5-sub-1

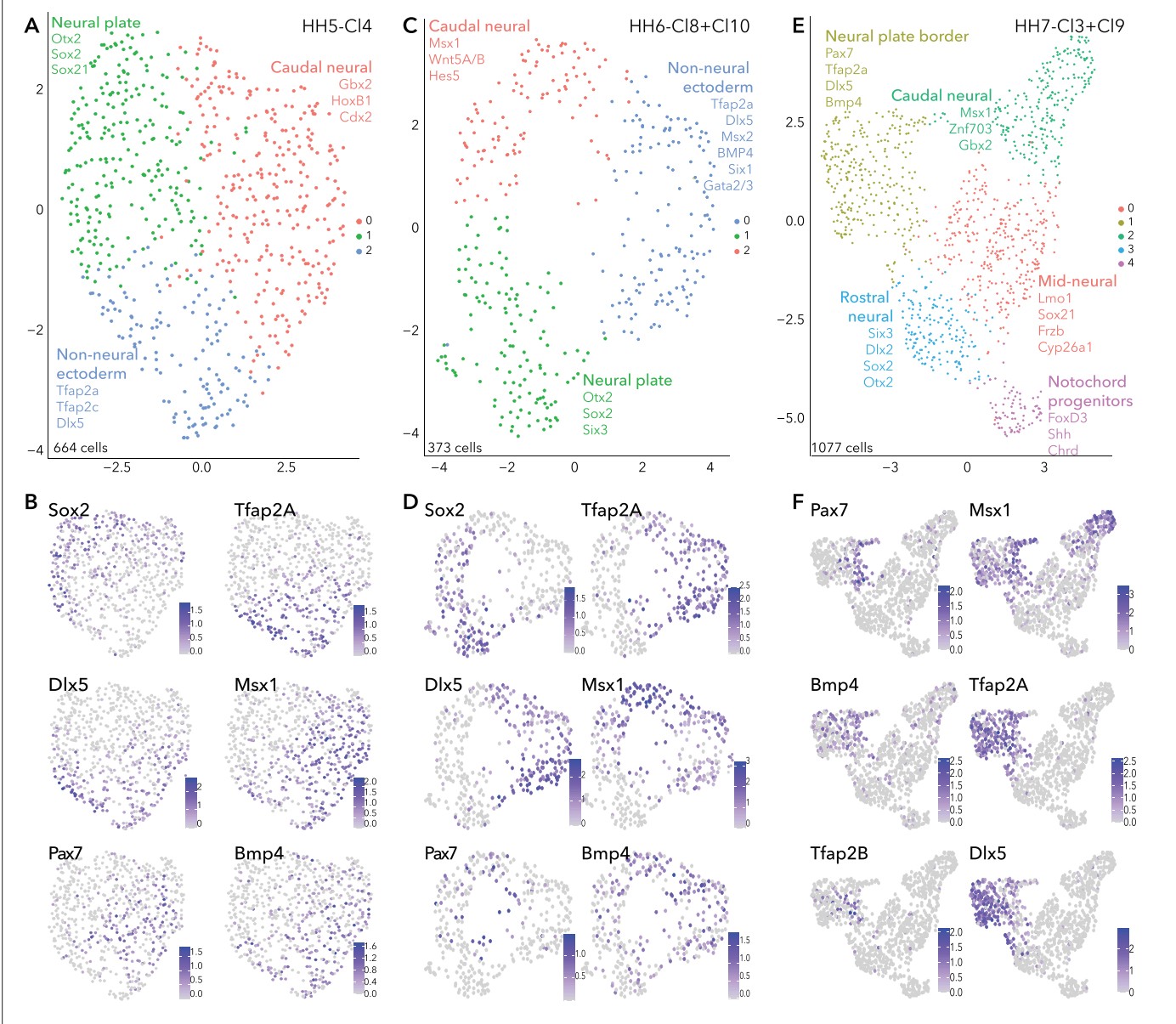

**Figure 2.** Subclustering ectoderm clusters extracted from whole epiblast data. (**A**) UMAP plot depicting three clusters resolved from HH5-Cl4. (**B**) Feature plots of selected genes in HH5 subclusters. (**C**) UMAP plot depicting three clusters resolved from HH6-Cl8 and HH6-Cl10. (**D**) Feature plots of selected genes in HH6 subclusters. (**E**) UMAP plot depicting five clusters resolved from HH7-Cl3 and HH7-Cl9. (**F**) Feature plots of selected genes in HH7 subclusters.

The online version of this article includes the following figure supplement(s) for figure 2:

**Figure supplement 1.** Supporting data for subclustering HH5, HH6, and HH7 ectoderm clusters.

(neural) but was clearly enriched in HH6-sub-1 (neural) (**Figure 2—figure supplement 1**). *Gbx2* was broadly expressed across caudal epiblast and non-neural ectoderm subclusters (HH5-sub-0, HH5-sub-2 and HH6-sub-0, HH6-sub-2), while *Znf703* was restricted to more caudal epiblast cells (HH5-sub-0 and HH6-sub-2) (**Figure 2—figure supplement 1**). *Gata2* and *Gata3* were found in a portion of HH5-sub-2 cells but their expression was expanded across HH6-sub-0 (**Figure 2—figure supplement 1**). The placode marker *Six1* was found at low levels across all subclusters at HH5, but at HH6, *Six1* was restricted to only a portion of HH6-sub-0, potentially to cells within a placode progenitor niche (**Figure 2—figure supplement 1**).

At HH7, ectodermal cells formed five discrete clusters (*Figure 2E*), suggesting lineage segregation was ongoing at this stage. Here neural plate border markers *Pax7*, *Msx1* and *Bmp4* were found in a niche of cells within HH7-sub-1, which broadly expressed *Tfap2A* and other non-neural ectoderm markers (*Figure 2F*, *Figure 2—figure supplement 1*). Neural crest genes (*Tfap2B*, *Draxin*, *Snai2*) were arising in the *Pax7*+ domain (*Figure 2F*, *Figure 2—figure supplement 1*). HH7-sub-3 subcluster was characterized by neural markers; *Six3*, *Otx2*, *Sox21*, and *Sox2* (*Figure 2—figure supplement 1*). Some HH7-sub-1 cells also expressed placode markers (*Dlx5*, *Pax6*, *Six1/3*) but these cells did not co-express *Pax7* (*Figure 2F*; *Figure 2—figure supplement 1*). Importantly, this reflects segregation of the neural plate border into medial (*Pax7*+) and lateral (*Tfap2A*+) regions, as well as highlighting the overlap and combination of genes co-expressed across these regions. HH7-sub-1 also featured *Irf6* expression, and *Grhl3* was present in a subset of cells which co-expressed *Irf6* and *Tfap2A* (*Figure 2—figure supplement 1*). Another subset of HH7-sub-1 cells expressed *Pax6*, possibly already delineating prospective lens placode at this stage (*Figure 2—figure supplement 1*), as these cells co-expressed *Six3*, known to activate *Pax6* by repressing Wnt signaling (*Liu et al., 2006*). HH7-sub-0 and HH7-sub-2 were also characterized as neural plate subclusters and could be resolved by axial markers, whereby mid-neural plate markers *Lmo1* and *Nkx6.2* were found in HH7-sub-0 and more caudally expressed genes *Cdx2/4*, *Hes5*, *Znf703* were found in HH7-sub-2 (*Figure 2—figure supplement 1*). Cells in HH7-sub-4 were enriched for *Shh*, *Chrd,* and *FoxD3*, suggesting they were notochord progenitors (*Figure 2—figure supplement 1*).

In summary, analysis of our single cell HH5 and HH6 transcriptomes did not identify a distinct cell cluster marked by known neural plate border genes; rather cells expressing these markers were distributed across neural and non-neural clusters. Our analysis reveals the complex heterogeneity of the developing neural plate border and shows neural plate border cells are not distinct as a transcriptionally separate group of cells until HH7 when they also begin to segregate into presumptive medial and lateral territories. Moreover, many factors are shared between the neural plate border and other ectodermal cells as the neural plate border is progressively established from the neural plate and non-neural ectoderm.

## Validation of gene expression using hybridization chain reaction reveals dynamic in vivo expression patterns of novel neural plate border genes

We validated the in vivo expression pattern of intriguing genes identified in our single-cell datasets, using fluorescent in situ hybridization (Hybridization Chain Reaction, HCR) (*Choi et al., 2018*), which enables simultaneous expression analysis of multiple transcripts. We used neural plate border markers (*Tfap2A*, *Pax7*, *Msx1*) to observe co-expression in this region. Embryos from HH4 through HH10 were screened to glean the time course of gene expression.

We identified several new genes in the ectoderm, many of which persisted into neural or non-neural tissues. At HH4 the chromatin remodeler, *Ing5*, Astacin-like metalloendopeptidase, *Astl*, and neuronal navigator 2, *Nav2* were enriched in the ectoderm cluster (HH4-Cl0) (*Figure 3A*, *Figure 3—figure supplement 1*). At stages HH4-HH6 *Ing5* was expressed across the neural plate border, whereas *Astl* expression was predominantly detected in the posterior neural plate border (*Figure 3B*, *Figure 3—figure supplement 1*). While both genes overlapped with *Tfap2A*, *Ing5* expression spread into the neural plate whereas *Astl* expression extended more laterally into the non-neural ectoderm (*Figure 3B'*). At HH8- *Astl* expression continued in the developing neural plate border and surrounding non-neural ectoderm, and also the neural plate (*Figure 3C and C'*). *Ing5* was also detected in the neural plate border and in the neural folds (*Figure 3C*). By HH8, *Astl* expression was prominent in the neural folds, most strongly in the posterior hindbrain (*Figure 3D and D'*). *Ing5* was detected along the neural tube at HH8 (*Figure 3D and D'*); however, by HH10 *Ing5* transcripts were no longer detectable. *Astl* expression was also broadly decreased at HH10, where activity was restricted to a small region of the neural tube in the hindbrain and the emerging otic placodes (*Figure 3—figure supplement 1*). Consistent with these observations *Astl* and *Ing5* were detected in HH5-Cl4 and HH6-Cl10 (*Astl*)/HH6-Cl8 (*Ing5*). *Astl* was seen in HH7-Cl9, *Ing5* was not differentially expressed at this stage but was present in HH7-Cl3 (*Figure 3—figure supplement 1*).

*Nav2,* detected in ectoderm clusters HH4-Cl0, HH5-Cl4, HH6-Cl10, and HH7-Cl9 was expressed across the ectoderm and neural plate at HH5–HH7, where it over-lapped with the neural marker *Sox21* (*Figure 3—figure supplement 1*), continuing into the neural folds at HH8 where *Nav2* was particularly

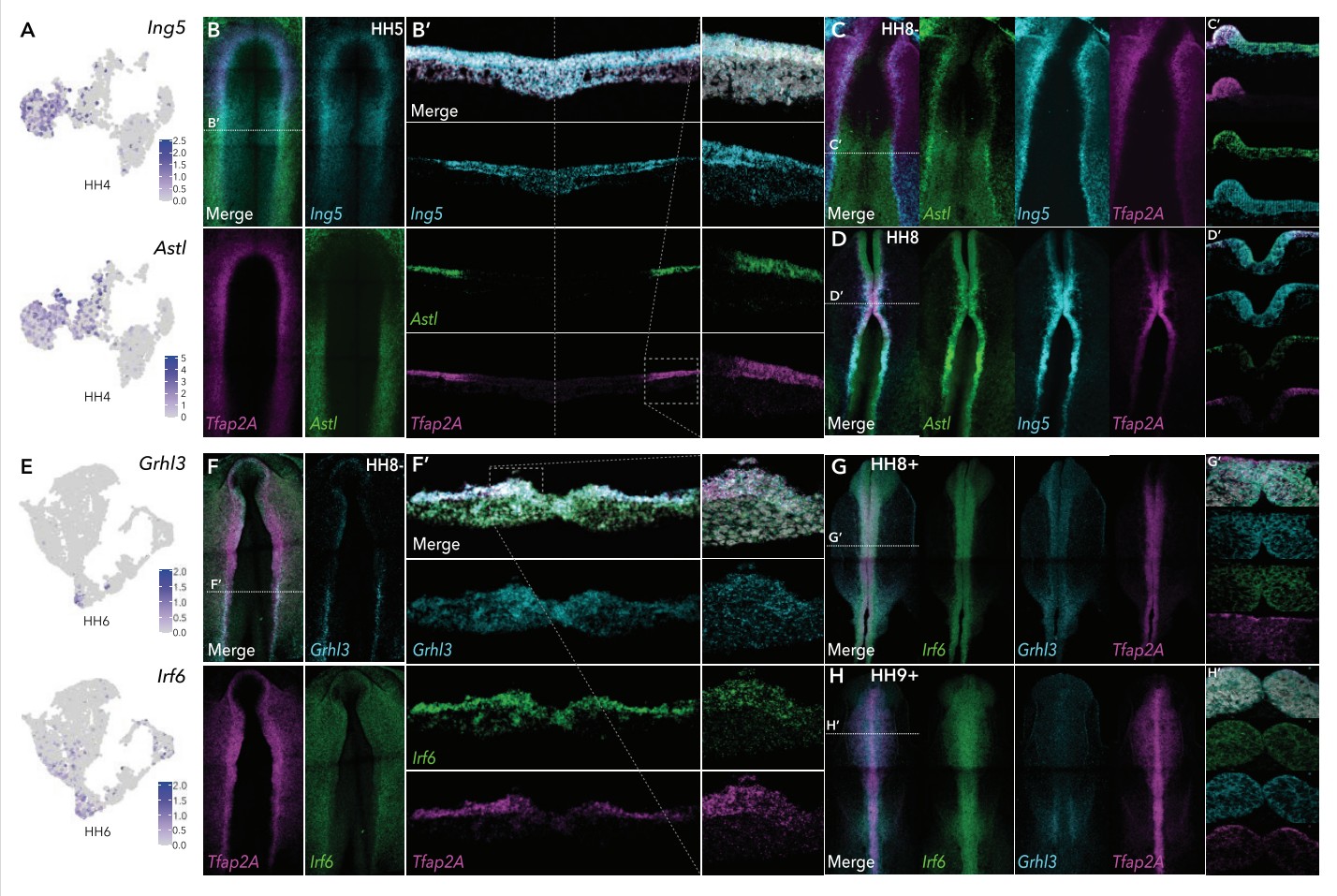

**Figure 3.** HCR validation of novel genes identified from whole epiblast data. (**A**) Feature plots of *Ing5* and *Astl* in HH4 data. (**B**) Whole mount HCR shows co-expression of *Ing5* and *Astl* with neural plate border marker *Tfap2A* at HH5. (**B'**) Transverse cryosection of (**B**) (dashed line). Dashed boxed region shown at high magnification in right panel. (**C–D**) Whole mount HCR shows co-expression of *Ing5* and *Astl* with neural plate border marker *Tfap2A* at HH8- (**C**) and HH8 (**D**). (**C'-D'**) Transverse cryosections of (**C**) and (**D**), respectively. (**E**) Feature plots of *Grhl3* and *Irf6* in HH6 data. (**F**) Whole mount HCR shows co-expression of *Grhl3* and *Irf6* with neural plate border marker *Tfap2A* at HH8-. (**F'**) Transverse cryosection of (**F**) (dashed line). Dashed boxed region shown at high magnification in right panel. (**G–H**) Whole mount HCR shows co-expression of *Grhl3* and *Irf6* with neural plate border marker *Tfap2A* at HH8+ (**G**) and HH9+ (**H**). (**G'-H'**) Transverse cryosections of (**G**) and (**H**), respectively.

The online version of this article includes the following figure supplement(s) for figure 3:

**Figure supplement 1.** HCR validation of selected genes.

prominent in the hindbrain (*Figure 3—figure supplement 1*). At HH9 *Nav2* expression was restricted to rhombomeres 2 and 4, whereas *Sox21* was more broadly expressed along the neural tube axis (*Figure 3—figure supplement 1*).

We identified a number of transcription factors in HH6-Cl8 (neural) and HH6-Cl10 (non-neural ectoderm). The latter was characterized by *Tfap2A* enrichment and also harbored *Grhl3* and *Irf6* (*Figure 3E*). In situ analysis showed *Grhl3* was specifically expressed in the neural plate border, most strongly in the posterior region at HH6 and HH8- (*Figure 3F*, *Figure 3—figure supplement 1*). *Irf6* was expressed in the neural plate border as well as the surrounding non-neural ectoderm at HH6 and HH8- (*Figure 3F*, *Figure 3—figure supplement 1*). Furthermore, we observed significant cellular co-expression of all three factors in the neural plate border (*Figure 3F'*). At HH8+, *Irf6* was restricted to the dorsal neural tube including premigratory neural crest cells, *Grhl3* was also detected here albeit at lower observable levels but was present in the emerging otic placode (*Figure 3G*). By HH9+ *Grhl3* levels were broadly diminished but remained in the developing otic placode. *Irf6* was maintained in the neural tube during neural crest delamination and also enriched in the otic placode (*Figure 3H*).

In the ectodermal subclusters, we identified a number of Wnt pathway genes. In HH6-sub-2, for example, we identified enrichment of Wnt signaling ligands *Wnt5A, Wnt5B, Wnt8A* (***Figure 2—figure supplement 1***) as well as Wnt processing factors *Sp5* and *Sp8* (***Figure 4A/E***). Conversely Wnt antagonists *Sfrp1, Sfrp2*, and *Frzb* were enriched in HH6-sub-1 (***Figure 2—figure supplement 1***). *Sp8* was identified in HH6-sub-2 (caudal neural), in vivo we found *Sp8* expression commenced from HH7 in the neural plate border, partially overlapping with *Tfap2A* (***Figure 4B***). The homeobox transcription factor, *Dlx6*, was present in HH7-sub-1, (neural plate) (***Figure 4A***) and was expressed predominantly in the anterior neural plate border or pre-placodal region (***Figure 4B***) as previously shown (***Anderson et al., 2016***). This expression pattern of both *Sp8* and *Dlx6* continued at HH8- (***Figure 4C***). *Sp8* expression was restricted to the dorsal neural tube, whereas *Dlx6* spread more laterally (***Figure 4C'***). At HH9, *Sp8* was detected predominantly in the anterior neural tube but was also found in premigratory neural crest cells in the mid-brain region (***Figure 4D and D'***) but largely absent from the hindbrain region, though some expression was seen in the trunk premigratory neural crest (***Figure 4D***). *Dlx6* was also present in premigratory neural crest cells at HH9 (***Figure 4D'***), as well as in the lateral non-neural ectoderm (***Figure 4D***).

*Sp5* was enriched in HH6-sub-2 (***Figure 4E***) and HH7-sub-2 (***Figure 2—figure supplement 1***). *Sp5* was detected in the anterior neural plate border where it overlapped with *Pax7* and *Msx1*, and in the posterior primitive streak/early mesoderm at HH7 (***Figure 4F/F'***). Later, (HH8-HH9) *Sp5* transcripts were restricted to the neural tube (***Figure 4G/H***) including premigratory neural crest cells (***Figure 4G'/H'***). At HH9 we also saw the onset of expression in the developing otic placodes (***Figure 4H***).

At HH6, HH6-Cl8 and HH6-Cl10 harbored *Znf703* and *Gbx2*, both of which were enriched in HH6-sub-2 (***Figure 4I***). *Znf703* has recently been demonstrated as RAR responsive factor important for neural crest development in *Xenopus* (***Hong and Saint-Jeannet, 2017***; ***Janesick et al., 2019***). Consistent with this, we found *Znf703* expression in the posterior neural plate extending into the neural plate border from HH6 (***Figure 4J/K***). At HH8, *Znf703* transcripts also populated the posterior neural folds (***Figure 4L***) and partially colocalized with *Msx1* in premigratory neural crest cells (***Figure 4L'***). We found *Gbx2* broadly expressed across the epiblast from HH6 (***Figure 4J/K***) consistent with previous reports (***Paxton et al., 2010***; ***Shamim and Mason, 1998***), becoming more enriched in the neural folds by HH8 (***Figure 4L***). While both *Znf703* and *Gbx2* partially localized with *Msx1*, *Znf703* expression continued medially into the neural plate whereas *Gbx2* transcripts extended laterally into the non-neural ectoderm (***Figure 4L'***).

Neural clusters contained numerous transcription factors including *Sox21* which was which was present in HH7-sub-0 and HH7-sub-3 (***Figure 2—figure supplement 1***). *Fezf2* together with *Otx2* and *Sox2* were also enriched in neural subclusters HH7-sub-3 (***Figure 2—figure supplement 1***). *Fezf2* has previously been shown to regulate *Xenopus* neurogenesis by inhibiting Lhx2/9 mediated Wnt signaling (***Zhang et al., 2014***) Accordingly, in vivo, Fezf2 was expressed in the anterior neural folds at HH7 where it overlapped with *Otx2*, and in the neural tube at later stages (***Figure 3—figure supplement 1***). *Lmo1* was also found in HH7-sub-0 (***Figure 2—figure supplement 1***) and was expressed across the neural plate at HH6 through HH8, where it was also seen in the neural folds (***Figure 3—figure supplement 1***).

## scVelo infers developmental trajectories from presumptive ectoderm to premigratory neural crest cells

The precise time at which the neural plate border segregates into the neural crest, placode and neural lineages has not been previously ascertained. To address this question, we used scVelo (***Bergen et al., 2020***) to resolve gene-specific transcriptional dynamics of these embryonic populations across time. This method is derived from RNA velocity (***La Manno et al., 2018***), which measures the ratio of spliced and unspliced transcripts and uses this information to predict future cell states, incorporating a critical notion of latent time that allows reconstruction of the temporal sequence of transcriptional steps. By generalizing the RNA velocities through dynamic modelling, this method is ideally suited to our analysis, as it reflects transitions in progressing, non-stationary embryonic populations such as neural crest, placode and neural plate progenitors under study here.

To assess the progression of ectodermal progenitors across time-points from epiblast to onset of neural crest emigration, we first integrated the ectoderm clusters from each stage namely; HH5-Cl4, HH6-Cl3 + 9 and HH7-Cl8 + 10. Here, we also included 10 X scRNA-seq data obtained from

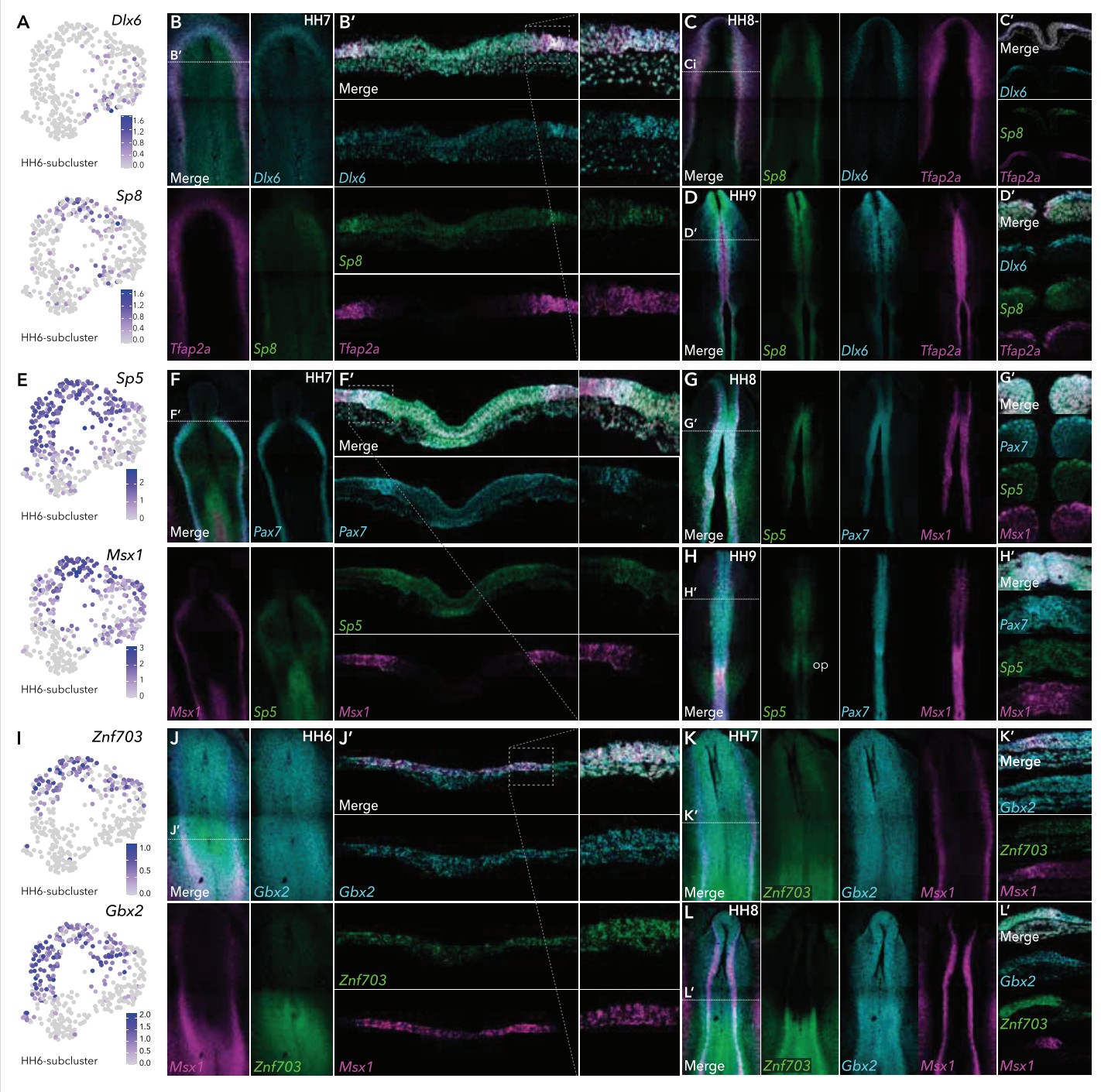

**Figure 4.** HCR validation of novel genes identified from extracted ectoderm clusters. (**A**) Feature plots of *Dlx6* and *Sp8* in HH6 subclusters. (**B**) Whole mount HCR shows co-expression of *Dlx6* and *Sp8* with neural plate border marker *Tfap2A* at HH7. (**B'**) Transverse cryosection of (**B**) (dashed line). Dashed boxed region shown at high magnification in right panel. (**C–D**) Whole mount HCR shows co-expression of *Dlx6* and *Sp8* with neural plate border marker *Tfap2A* at HH8- (**C**) and HH9 (**D**). (**C'-D'**) Transverse cryosections of (**C**) and (**D**), respectively. (**E**) Feature plots of *Sp5* and *Msx1* in HH6 subclusters. (**F**) Whole mount HCR shows co-expression of *Sp5* with neural plate border markers *Msx1* and *Pax7* at HH7. (**F'**) Transverse cryosection of (**E**) (dashed line). Dashed boxed region shown at high magnification in right panel. (**G–H**) Whole mount HCR shows co-expression of *Sp5* with neural plate border markers *Msx1* and *Pax7* at HH8 (**G**) and HH9 (**H**). (**G'-H'**) Transverse cryosections of (**G**) and (**H**) respectively. (**I**) Feature plots of *Znf703* and *Gbx2* in HH6 subclusters. (**J**) Whole mount HCR shows co-expression of *Znf703* and *Gbx2* with neural plate border marker *Msx1* at HH6. (**J'**) Transverse cryosection of (**J**) (dashed line). Dashed boxed region shown at high magnification in right panel. (**K–L**) Whole mount HCR shows co-expression of *Znf703* and *Gbx2* with neural plate border marker *Msx1* at HH7 (**K**) and HH8 (**L**). (**K'-L'**) Transverse cryosections of (**K**) and (**L**) respectively. Op; otic placode.

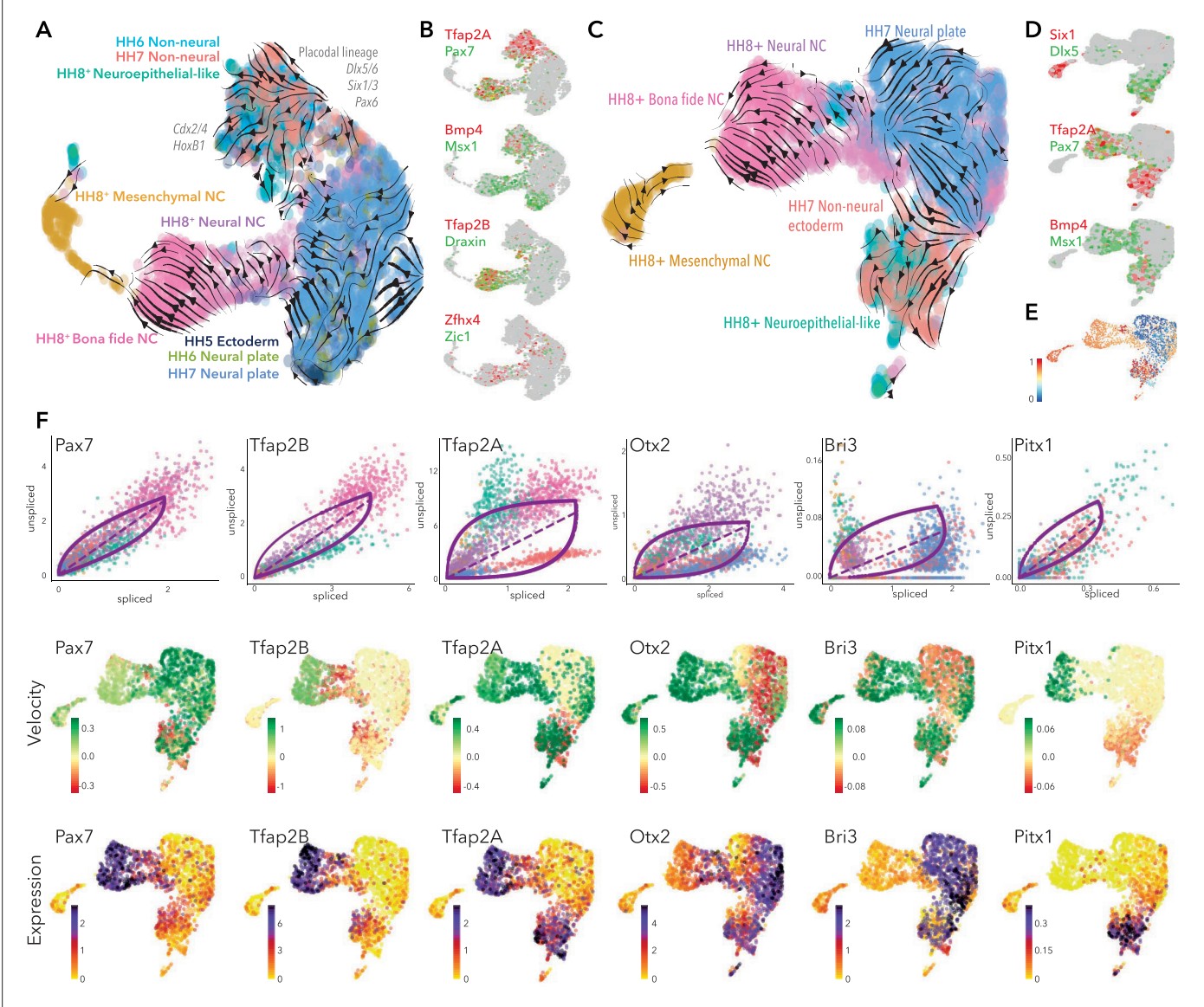

**Figure 5.** Developmental trajectory predictions from the neural plate border to premigratory neural crest. (**A**) UMAP embedding of ectoderm clusters combined with premigratory neural crest data showing predicted trajectories. (**B**) Colocalization of selected genes across the combined data set. (**C**) UMAP embedding of HH7 ectoderm clusters combined with premigratory neural crest data showing predicted trajectories. (**D**) Colocalization of selected genes across HH7/premigratory neural crest combined data set. (**E**) Latent time data from HH7/premigratory neural crest combined data-set. Scale represents oldest (0, blue) to newest cells (1, red). (**F**) Phase portraits (top row) colors correlate with those in A/C, velocity plots (middle row) and expression plots (bottom row) of selected genes across the HH7/ premigratory neural crest combined data sets. NC; neural crest.

The online version of this article includes the following figure supplement(s) for figure 5:

**Figure supplement 1.** Supporting data for combined data set scVelo analysis.

**Figure supplement 2.** scVelo analysis at individual time points.

premigratory neural crest cells (HH8+) (*Williams et al., 2019*), such that we could trace trajectories from the neural plate border to the neural crest. As well as identifying bona fide neural crest cells (*FoxD3, Tfap2B, Sox10*), this work also defined distinct clusters of neural crest cells characterized by marker genes representing their likely potential; neural-neural crest (*Sox3, Pax6, Cdh2*) and mesenchymal-neural crest (*Pitx2, Twist1, Sema5B*).

Combining these datasets revealed that the non-neural ectoderm from both HH6 and HH7 grouped together, as did the neural plate clusters from the same stages (*Figure 5A*, *Figure 5—figure supplement 1*). This suggests successful inter-sample incorporation into a single reference with no

resultant batch bias. Ectoderm cells from HH5 were spread amongst the neural plate and the non-neural ectoderm clusters from HH6 and HH7. The neural crest clusters were more discrete, with the bona fide neural crest cluster clearly segregated from neural and mesenchymal neural crest clusters as previously described (*Williams et al., 2019*). The HH8+ dataset also included a neuroepithelial-like cluster, characterized by *Cdh1, Epcam,* and *FoxG1* expression. Interestingly these cells grouped with the non-neural ectoderm clusters from HH6 and HH7, including neural plate border cells (*Pax7, Tfap2A*; *Figure 5A/B*), suggesting early neural crest cells retain properties of their predecessors. scVelo predicted that non-neural ectoderm cells from HH6/HH7 were split between two trajectories (*Figure 5A*). Placodal markers (*Dlx5/6, Six3,* and *Pax6*) defined one distinct trajectory, whereas the other trajectory was less well defined but could be distinguished by posterior epiblast markers (*HoxB1, Cdx4*) (*Figure 5—figure supplement 1*). The neuroepithelial-like cells from HH8+ were emerging between these trajectories where they were joined by other cells from HH6/HH7 non-neural ectoderm clusters. These cells were enriched for *Pax7* and other neural plate border markers *Tfap2A, Msx1, Bmp4* (*Figure 5B*). Early neural crest genes including *Tfap2B* and *Draxin* were also emerging here (*Figure 5B*). While these cells had less defined trajectories, they were generally directed toward the neural-neural crest lineage (*Figure 5A*). Neural plate border and neural crest markers also extended into a heterogeneous region of cells, where a significant portion of the neural-neural crest and neural plate cells were highly integrated, suggesting some cells from these populations shared transcriptional signatures. This region was further defined by *Zfhx4* and *Zic1* (*Figure 5B*) which are expressed in the neural plate border, neural plate and premigratory neural crest cells (*Khudyakov and Bronner-Fraser, 2009*; *Williams et al., 2019*). Some neural-neural crest cells within this region appeared to be directly derived from the neural plate.

Since we did not see evidence of neural crest lineages until HH7, we assessed the developmental trajectories from HH7 ectoderm clusters directly to bona fide neural crest cells (HH8). To this end, we extracted the ectoderm clusters (HH7-Cl3+ Cl9) and combined these cells with the premigratory neural crest data set from HH8+ embryos (*Figure 5—figure supplement 1*) and used scVelo to predict the trajectories across these cells/stages. Analysis of the pooled dataset yielded clearly defined putative developmental trajectories splitting from the HH7 ectoderm clusters, whereby the non-neural ectoderm cells (HH7-Cl9) appeared to give rise to the neuroepithelial-like cells from the HH8+ dataset, and the neural plate (HH7-Cl3) cells were contributing directly to the neural-neural crest (*Figure 5C*). Non-neural ectoderm cells were also initiating a separate trajectory, likely representing placodal lineages as indicated by *Six1, Dlx5* (*Figure 5D*). While *Tfap2A* was expressed across the non-neural ectoderm cluster, *Pax7* was found in just a subset of non-neural ectoderm cells and in neuroepithelial-like cells, where *Tfap2A* was also found. Both factors were also enriched in *bona fide* neural crest, though *Pax7* was also detected in the neural-neural crest (*Figure 5D*). Other neural plate border markers (*Bmp4, Msx1*) were found in the non-neural ectoderm and some neuroepithelial-like cells, as well as extending into the interface of neural plate and neural-neural crest populations (*Figure 5D*).

Taken together, combining our 10X data-sets from all stages reinforced the notion that the neural plate border territory is not specified to a neural crest fate until HH7 since scVelo analysis on the entire dataset (including HH5 and HH6 data) generated ambiguous and non-contiguous developmental trajectories (*Figure 5A*). While neural crest trajectories were not clearly defined until HH7, placodal trajectories could be discerned from HH6. Latent time analysis supported these observations, indicating the progression of transcriptional maturity and corresponding differentiation status across the HH7 non-neural ectoderm cells into the neuroepithelial-like cells and, similarly, from the neural plate cells into the neural-neural crest and canonical neural crest populations (*Figure 5E*). Furthermore, the data showed a sub-population of premigratory neural crest cells shared transcriptional signatures with their progenitor cells.

## scVelo analysis infers developmental trajectories across the epiblast

We also examined predicted trajectories across the whole embryo data from individual time points (*Figure 5—figure supplement 2*). At HH4 we observed segregation of the germ layers with all ectoderm cells (HH4-Cl0) aligning along the same trajectory. Epiblast stem cells (HH4-Cl2) were contributing to all other clusters. At HH5, the majority of ectoderm cells (HH5-Cl4) still followed the same trajectory. A clear split in the ectoderm populations was first observed at HH6 with neural plate (HH6-Cl8)

and non-neural ectoderm (HH6-Cl10) initiating separate trajectories. This divergence continued at HH7 with the neural plate (HH7-Cl3) and non-neural ectoderm (HH7-Cl9) cells following separate trajectories (*Figure 5—figure supplement 2*). However, there was also some convergent trajectories between these clusters where neural plate border, ectodermal ,and placode genes (*Pax7*, *Bmp4*, *Tfap2A*, *Msx1*, *Dlx5,* and Six3) were co-expressed (*Figure 5—figure supplement 2*), suggesting that some cells are not yet lineage-restricted but, rather, remain an intermingled heterogeneous progenitor population. Meanwhile, other cells from HH7-Cl3 and HH7-Cl9 were linking to the neural plate/ tube trajectories (HH7-Cl4). We observed a portion of neural plate (HH7-Cl3) cells joining the posterior mesoderm cluster (HH7-Cl0), likely containing neuromesodermal precursors (NMPs) as suggested by the enrichment in *Tbxt* and *Sox2* expression (*Figure 5—figure supplement 2*).

## Neural plate border lineages emerge progressively from the ectoderm over time

To further resolve the ectodermal trajectories, the extracted ectoderm clusters were modelled using scVelo. At HH5, each population (HH5-sub-0, caudal epiblast; HH5-sub-1, neural plate; HH5-sub-2, non-neural ectoderm) initiated a separate trajectory. However, cells located centrally, including neural plate border cells, contributed to all three subclusters (*Figure 5—figure supplement 2*).

At HH6 the trajectories were more distinct. Caudal epiblast cells (HH6-sub-2) contributed to both neural plate and non-neural ectoderm populations. Within the neural plate cluster (HH6-sub-1) some cells were directed toward the non-neural ectoderm and others followed a separate trajectory. Likewise, the non-neural ectoderm cluster was subdivided between two parallel pathways with some mutual cross-over. Neural plate border cells found across all three clusters and at their interface largely joined the non-neural ectoderm clusters (*Figure 5—figure supplement 2*).

HH7 trajectory map was more complex (*Figure 5—figure supplement 2*). Here *Pax7*+ neural plate border cells (HH7-sub-1, *Figure 2F*) appeared to be contributing to both putative placode and neural crest progenitors, emerging within HH7-sub-Cl1, as well as projecting toward neural clusters (HH7-sub-0). Some neural plate border cells/neural crest progenitors appeared to be derived from the neural plate (HH7-sub-0/3). This suggests that *Pax7*+ cells are capable of giving rise to multiple lineages and highlights heterogeneous combinations of factors at the neural plate border reflect multipotency. The remainder of HH7-sub-1 cells (*Pax7*-) were engaged in a separate trajectory; these cells were enriched for *Dlx5* (*Figure 2F*), *Six1*, *Pax6,* and *Gata3* (*Figure 2—figure supplement 1*), suggesting a placodal fate. Overall, at HH7 scVelo analysis showed the neural plate border was beginning to segregate according to progenitor populations.

## Analysis of dynamic transcriptional trajectories during lineage restriction reveals putative lineage-specific drivers

We next sought to identify candidate genes putatively driving the observed developmental trajectories in the context of neural crest specification from the neural plate border. Dynamical modeling of transcriptional states within the HH7/8 combined data-set revealed a number of highly ranked dynamic genes including *Pax7* and *Tfap2B*. *Pax7* splicing increased progressively from the non-neural ectoderm cells to the neuroepithelial-like cells and lower splicing was observed in neural crest cells where the transcripts stabilized such that increased expression was observed (*Figure 5F*). Splicing of *Pax7* transcripts was also increased in the putative placodal lineage from the non-neural ectoderm (*Figure 5F*). *Tfap2B* splicing was not evident in the HH7 data, but was increased in the neuroepithelial-like cells and stabilized in neural crest populations (*Figure 5F*). *Tfap2A* showed more complex velocities, whereby high levels of spliced transcripts were detected in non-neural ectoderm cells from HH7 likely driving these cells toward the placode trajectory (*Figure 5F*). We also observed a progressive increase in *Tfap2A* splicing across the neuroepithelial-like cells to the *bona fide* neural crest. This suggested that by HH8, *Tfap2A* transcripts are stable in the neuroepithelial-like population but upregulated in the bona fide neural crest, consistent with the observed expression dynamics (*Figure 5F*). Another neural plate border gene, *Bmp4*, was dynamically regulated in non-neural ectoderm but downregulated in the neural plate (*Figure 5—figure supplement 1*).

We also identified a number of more novel factors putatively involved in ectoderm lineage progression. *Otx2* was driving neural plate cells toward the neural-neural crest cluster, consistent with previous findings from functional perturbation studies (*Williams et al., 2019*). However, *Otx2* also seemed to

be driving non-neural ectoderm cells toward the neuroepithelial-like population (*Figure 5F*). *Lmo1* was also driving the neural plate cells toward the neural-neural crest; however, unlike *Otx2*, *Lmo1* expression was not maintained in the neural crest populations (*Figure 5—figure supplement 1*). *Bri3* was identified as a highly ranked dynamic gene and was upregulated in neural plate cells and some non-neural ectoderm cells. Down-regulation of *Bri3* in neural crest cells suggested an early role in ectoderm lineage trajectories for this novel factor (*Figure 5F*). *Nav2* and *Sox11* were both highly expressed in the neural plate and downregulated in the non-neural ectoderm. *Sox11* was also expressed in the mesenchymal neural crest cluster where it was highly spliced, potentially representing a dual segregated role for this factor in both neural plate and mesenchymal crest lineages (*Figure 5—figure supplement 1*). *Nav2* velocities increased progressively from the non-neural ectoderm to the neural plate; consistently, a subset of non-neural ectoderm cells joined a trajectory with the neural plate cells (*Figure 5F*) where *Nav2* splicing was highest (*Figure 5—figure supplement 1*). We found *Pitx1* and *Dlx6* to be dynamically active across the merged HH7/8 data-set, potentially driving non-neural ectoderm to the neuroepithelial-like population (*Figure 5F*, *Figure 5—figure supplement 1*).

This analysis provides insight into potential candidates driving lineage specific circuits underlying the progressive segregation of the early ectoderm into neural and non-neural ectoderm and concomitantly initiating the emergence of early neural crest and placode progenitors.

## Discussion

Here, we use single-cell RNA-sequencing of chick embryos from late gastrula through early neurula to characterize the development of the neural plate border and its derivatives, the neural crest and cranial placode precursors. Our data show that the neural plate border, as defined by co-expression of *Tfap2A* and *Pax7* first emerges at HH5, but is not fully transcriptionally defined until HH7. Previous work has pointed to the presence of a pre-border region at blastula stages harboring neural crest progenitors demarcated by *Pax7* expression (*Basch et al., 2006*; *Prasad et al., 2020*). However, this was observed in explanted cultures. In contrast, we do not detect significant *Pax7* expression until HH5. This suggests that cells within the explants may not have been fully specified at the time of explantation but cell interactions coupled with autonomous programming enabled the cells to continue their specification program. This highlights the importance of examining tissue dynamics within their endogenous context.

We identified numerous genes in our dataset that have not previously been shown to be expressed the in developing neural plate border, several of which can be further placed in the broader neural crest gene regulatory network. Using the ShinyApp associated with early neural crest RNA-seq data (*Williams et al., 2019*) we found *Astl* expression correlated with neural crest specifier genes including *Msx1* and *Snai2* (WGCNA cluster-*i*). Whereas *Ing5* expression correlated with *Pitx1* (WGCNA cluster-*x*), which we identified here as a putative driver of neuro-epithelial neural crest lineage. Another example is *Irf6* which has previously been described in craniofacial development (*Fakhouri et al., 2017*; *Ingraham et al., 2006*; *Wang et al., 2003*) but has yet to be explored at earlier stages of development. *Irf6* has been suggested to activate *Grhl3* in the zebrafish periderm (*de la Garza et al., 2013*) and mutations in *Irf6* or *Grhl3* are associated with neural crest defects (cleft-lip/palate) characteristic of Van der Woude syndrome (*Peyrard-Janvid et al., 2014*). Furthermore, *Grhl3* has been suggested to work in a module with *Tfap2A* and *Irf6* during neurulation (*Kousa et al., 2019*). We found *Grhl3* putatively regulates *Tfap2A/B*, *Gli2/3*, and *Sox10*, but is itself downregulated by *FoxD3* in early neural crest development (*Williams et al., 2019*), consistent with our HCR analysis where *Grhl3* expression is decreased in the premigratory neural crest. In addition, Wnt signaling pathway genes like *Sp5* and *Sp8* were prominent in our data-sets. Wnt signaling has a well-established role in neural plate border specification (*Pla and Monsoro-Burq, 2018*; *Schille and Schambony, 2017*); for example, Wnt signals are required to activate the earliest neural crest genes *Tfap2A*, *Gbx2* and *Msx1* (*de Crozé et al., 2011*; *Li et al., 2009*). *Sp8* is proposed to play a role in processing Wnt signals during limb development (*Haro et al., 2014*; *Kawakami et al., 2004*), but has also been described in neural patterning and craniofacial development (*Sahara et al., 2007*; *Zembrzycki et al., 2007*), whereby loss of *Sp8* in mice caused severe craniofacial defects due to increased apoptosis and decreased proliferation of neural crest cells (*Kasberg et al., 2013*). However, an early role for *Sp8* has not been explored. *Sp5* has recently been implicated in the neural crest gene regulatory network, where it was found to negatively regulate *Axud1* to help maintain neural crest cells in a naive state (*Azambuja and Simoes-Costa,*

*2021*). Therefore, *Sp5* and *Sp8* may represent hitherto unknown components of Wnt mediated induction of neural plate border lineages. *Gbx2* was recently reported as the earliest Wnt induced neural crest induction factor in *Xenopus*. Perturbation of *Gbx2* inhibited neural crest development while the placodal population was expanded (*Li et al., 2009*). Interestingly, our analysis reveals distinct heterogeneity of Wnt signaling factors, suggesting differential cellular responses to Wnt signaling within the developing neural plate border, which may contribute to driving different lineage trajectories.

While these datasets reveal a number of intriguing genes, many were not exclusively expressed in the neural plate border. This demonstrates that the neural plate border is not distinguishable by a unique set of genes, but rather it is the combination of genes shared across this region and surrounding tissues that endows its unique multipotency. This is consistent with and expands upon previous results revealing colocalization of markers characteristic of multiple lineages in the neural plate border (*Roellig et al., 2017*). Moreover, we find that factors associated with neural plate border derivatives are also expressed in the emerging neural plate border, raising the intriguing possibility that these factors may have a role in establishing neural plate border lineages.

We used scVelo analysis of transcriptional and splicing dynamics across clustered and aligned single-cell transcriptomes to predict the dynamics of developmental trajectories during ectoderm lineage segregation. This enabled us to follow the transcriptional velocity of individual genes and resolve their dynamics at the single-cell level. Significantly this provides a high-resolution temporal dimension to our understanding of the changing ontology of the neural plate border and its derivatives. The results suggest that the ectoderm is initiating the division of trajectories between neural and non-neural progenitors at HH5, but the neural plate border is not yet transcriptionally distinct. At HH6 and HH7, the neural plate border population is emerging at the interface of these tissues, from which multiple lineages manifest by HH7. This analysis also shows neural-neural crest cells are the first to surface from the neural plate border and neural plate. From these, the bona fide neural crest emerges followed by mesenchymal neural crest. Furthermore, this analysis shows a subset of premigratory neural crest cells share signatures with their precursors from neural plate border cells from HH6 and HH7, demonstrating that some early neural crest cells are not yet restricted to a particular lineage.

Taken together our data show that cells of the emerging neural plate border are not characterized by unique transcriptional signatures, but share features with cells of the surrounding ectoderm. This highlights the heterogeneity of the neural plate border whereby *Pax7*+ cells are integrated with other ectoderm populations. While Pax7 is the predominant factor in the neural plate border and has been suggested to label neural crest precursors as early as HH5 (*Basch et al., 2006*), the complexity and co-expression signatures are what endow neural plate border cells with their unique multipotency compared to neural plate and ectoderm cells. Interestingly, we note that neural plate border signatures are not apparent until HH7, later than previously suggested. By contrast, placodal trajectories were discernible from HH6. Moreover, we uncover the possibility that *Pax7*+ cells give rise to all neural plate border lineages, but lose multipotency signatures at the onset of lineage specific trajectories. Our analysis provides important insights into genes underlying the progressive segregation of the emergence of early neural crest and placode progenitors at the neural plate border. By revealing lineage trajectories over developmental time, this resolves the timing of neural plate border lineage segregation whilst also informing on dynamics of multipotency programmes.

# Materials and methods

**Key resources table**

| Reagent type (species) or resource | Designation | Source or reference | Identifiers | Additional information |
|---|---|---|---|---|
| Gene (*Gallus gallus*) | Fertilized hens eggs | Sunstate Ranch Sylmar CA | | |
| Commercial assay or kit | Chromium Single Cell 3' v3 Library | 10 X Genomics | Cat. #1000075 | |
| Software, algorithm | Seurat v3 CellRanger v3.1.0 | *Stuart et al., 2019* Doi:10.1016/j.cell.2019.05.031 *Zheng et al., 2017* DOI: 10.1038/ncomms14049 | | |

## Chick embryos

Fertilized chicken eggs, obtained from Sunstate Ranch Sylmar CA, were incubated at 37°C with approximately 40% humidity. Embryos were staged according to *Hamburger and Hamilton, 1951*.

## Preparing embryos for FAC-sorting

Appropriately staged embryos were extracted using the filter paper based 'easy-culture' method. Eggs were opened after desired incubation period, albumin was removed and embryos were lifted from the yolk using punctured filter paper, this procedure is described in detail elsewhere (*Williams and Sauka-Spengler, 2021b*). Embryos were kept in Ringers solution and dissected to remove all extra-embryonic material. Embryos were then dissociated for fluorescence activated cell sorting (FACS) as previously described (*Williams and Sauka-Spengler, 2021a*). Embryos were processed by FACS using 7-AAD as a live/dead stain such that healthy individual cells were obtained with a reliable cell count.

## 10X single-cell RNA-Seq library preparation and sequencing

Approximately 10,000 cells/stage were collected by FACS into 2 µl Hanks buffer then loaded onto the 10 X Genomics Chromium platform. Single-cell RNA-seq libraries were generated using the Chromium Single Cell 3' Library and Gel Bead Kit v3 (10 X Genomics, Cat. #1000075) (HH4 data was obtained using v2 chemistry) as per the manufacturers protocol. Libraries were quantified using Qubit (Life Tech Qubit high sensitivity DNA kit Cat. #Q32854) and Kapa (Kapa Biosystems, KAPA Library Quantification Kit, Cat. #KK4835). scRNA-seq libraries were sequenced on Illumina HiSeq2500 platform in rapid run mode with on-board clustering and sequencing depth of 300 million reads. The run type was: paired end 28(read1)–8(index)–91(read2). HH4 10 X scRNA-seq library was sequenced on Illumina NextSeq500 platform using high output v2.5 150-cycle kit in 26 × 8 x 0 x 98 mode.

## 10X single-cell RNA-Seq data analysis

Fastq files were generated using cellranger v3.1.0 mkfastq (*Zheng et al., 2017*). Cellranger was also used for base calling, demultiplexing and mapping to the galgal6 genome assembly. A custom galgal6 genome was constructed using the mkref function whereby selected 3'UTRs were extended according to manually annotation.

From HH4 embryos 2398 cells were recovered with 144,605 mean reads/cell, 1836 median genes/cell, 6,180 median UMI counts/cell and 15,370 total genes. From HH5 embryos 6723 cells were recovered with 44,343 mean reads/cell, 3,659 median genes/cell, 14,229 median UMI counts/cell and 18,916 total genes. From HH6 embryos 4553 cells were recovered with 30,466 mean reads/cell, 3396 median genes/cell, 13,309 median UMI counts/cell and 17,828 total genes. From HH7 embryos 6585 cells were recovered with 20,044 mean reads/cell, 2696 median genes/cell, 8821 median UMI counts/cell and 17,873 total genes. Count matrices were generated using the cellranger count function and exported to R-studio for downstream analysis using Seurat-v3 (*Stuart et al., 2019*). Matrices were filtered to remove barcodes with fewer than 500 genes and more than 3500–5500 genes (Figure S1/S2) and high mitochondrial content ( > 0.5%). UMI counts were normalized and following principal component analysis linear dimension reduction was conducted (HH4; resolution 0.2, dims 1:20, HH5; resolution 0.4, dims 1:15, HH6; 0.6, dims 1:20, HH7; resolution 0.4, dims 1:20). Clustered cells were visualized on a UMAP plot and differentially expressed genes were identified. Ectoderm clusters were extracted and re-clustered using the 'subset' command, analysis was performed as for whole embryo data. scVelo analysis: scVelo dynamical modelling was performed in python (*Bergen et al., 2020*). Loom files containing spliced/unspliced transcript expression matrices were generated using velocyto.py pipeline (*La Manno et al., 2018*). "Loom cell" names were renamed to match Seurat object cell names and only Seurat-filtered cells were selected for trajectory analysis. Seurat generated UMAP coordinates, clusters and cluster colors were added to the filtered 'loom cells'.

## Hybridization chain reaction

Fluorescent in situ hybridization chain reaction was performed using the v3 protocol (*Choi et al., 2018*). Briefly, embryos were fixed in 4% paraformaldehyde (PFA) for 1 hr at room temperature, dehydrated in a methanol series and stored at –20 °C at least overnight. Following rehydration embryos were treated with Proteinase-K (20 mg/mL) for 1–2.5 min depending on stage (1 min HH4-6, 2.5 min

for older embryos) at room temperature and post-fixed with 4% PFA for 20 min at room temperature. Embryos were washed in PBST for 2 × 5 min on ice, then 50% PBST / 50% 5 X SSCT (5 X sodium chloride sodium citrate, 0.1% Tween-20) for 5 min on ice and 5 X SSCT alone on ice for 5 min. Embryos were then pre-hybridized in hybridization buffer for 5 min on ice, then for 30 min at 37 °C in fresh hybridization buffer. Probes were prepared at 4 pmol/mL (in hybridization buffer), pre-hybridization buffer was replaced with probe mixture and embryos were incubated overnight at 37 °C with gentle nutation. Excess probes were removed with probe wash buffer for 4 × 15 min at 37 °C. Embryos were pre-amplified in amplification buffer for 5 min at room temperature. Hairpins were prepared by snap-cooling 30 pmol (10 ml of 3 mM stock hairpin) individually at 95 °C for 90 s and cooled to room temperature for minimum 30 min, protected from light. Cooled hairpins were added to 500 µl amplification buffer. Pre-amplification buffer was removed from embryos and hairpin solution was added overnight at room temperature, protected from light. Excess hairpins were removed by washing in 5 X SSCT 2 × 5 min, 2 × 30 min and 1 × 5 min at room temperature. Embryos were mounted on slides and imaged using Zeiss LSM 880 Upright confocal microscope. Images were processed using Zeiss Zen software, Z-stacks scans were collected at 6 µm intervals across approximately 70–100 µm, maximum intensity projections of embryo z-stacks are presented. Tile scanning was used (2 × 3) and stitched using bidirectional stitching mode, with overlap of 10%. For sectioned samples, images were obtained on a Zeiss LSM 880 Upright confocal with 20 X and 40 X oil immersion objectives, single z-slices are shown.

## Cryosectioning

Following HCR and whole mount imaging, selected embryos were prepared for cyrosectioning Embryos were placed in a 15% sucrose solution at 4 °C overnight then 15% sucrose / 7.5% Gelatin overnight at 37 °C. Embryos were transferred to 20% gelatin and incubated at 37 °C for 4 hr then mounted in 20% gelatin, snap frozen in liquid nitrogen and stored at –80 °C. Sections were taken at 10 µm intervals.

## Acknowledgements

Fluorescence activated cell sorting was performed at California Institute of Technology Flow Cytometry Facility using BD Biosciences FACSAria Cell Sorter with Patrick Cannon. 10 X libraries were prepared in the Thomson lab at California Institute of Technology with assistance from Jeff Park. Illumina sequencing was performed at the Millard and Muriel Jacob at California Institute of Technology with Igor Antoshechkin. HH4 10 X library was constructed at MRC Weatherall Institute of Molecular Medicine, University of Oxford with assistance from Kevin Clark (FACS facility), Dr Neil Ashley (single-cell facility) and Tim Rostron (NGS sequencing facility). Confocal microscopy was performed within the Biological Imaging Facility at the Beckman Institute, California Institute of Technology with assistance from Dr Giada Spigolon. We thank members of the Bronner and Sauka-Spengler labs for their support and helpful discussions. This work was funded by NIH R01DE027538 to MEB/RMW, Wellcome Trust Senior Research Fellowship (215615/Z/19/Z) to TSS/RMW and Radcliffe Department of Medicine Scholarship and MRC DTP Supplementary Funding to ML.

## Additional information

### Competing interests

Marianne E Bronner: Senior editor, *eLife*. The other authors declare that no competing interests exist.

### Funding

| Funder | Grant reference number | Author |
| --- | --- | --- |
| National Institutes of Health | R01DE027538 | Marianne E Bronner<br>Ruth M Williams |
| Wellcome Trust | 215615/Z/19/Z | Tatjana Sauka-Spengler<br>Ruth M Williams |

| Funder | Grant reference number | Author |
| --- | --- | --- |
| Radcliffe Department of Medicine Scholarship | | Martyna Lukoseviciute |

The funders had no role in study design, data collection and interpretation, or the decision to submit the work for publication.

## Author contributions

Ruth M Williams, Conceptualization, Data curation, Formal analysis, Investigation, Methodology, Validation, Visualization, Writing – original draft; Martyna Lukoseviciute, Formal analysis, Writing – review and editing; Tatjana Sauka-Spengler, Formal analysis, Funding acquisition, Supervision, Writing – review and editing; Marianne E Bronner, Conceptualization, Funding acquisition, Project administration, Resources, Supervision, Writing – review and editing

## Author ORCIDs

Ruth M Williams http://orcid.org/0000-0002-2628-7834
Tatjana Sauka-Spengler http://orcid.org/0000-0001-9289-0263
Marianne E Bronner http://orcid.org/0000-0003-4274-1862

## Decision letter and Author response

Decision letter https://doi.org/10.7554/eLife.74464.sa1
Author response https://doi.org/10.7554/eLife.74464.sa2

## Additional files

### Supplementary files
- Transparent reporting form
- Source data 1. 10X summary.

### Data availability

Sequencing data have been deposited in GEO under accession codes GSE181577.

The following dataset was generated:

| Author(s) | Year | Dataset title | Dataset URL | Database and Identifier |
| --- | --- | --- | --- | --- |
| Williams RM, Lukoseviciute M, Sauka-Spengler T, Bronner M | 2021 | Segregation of neural crest specific lineage trajectories from a heterogeneous neural plate border territory only emerges at neurulation | https://www.ncbi.nlm.nih.gov/geo/query/acc.cgi?acc=GSE181577 | NCBI Gene Expression Omnibus, GSE181577 |

The following previously published datasets were used:

| Author(s) | Year | Dataset title | Dataset URL | Database and Identifier |
| --- | --- | --- | --- | --- |
| Williams RM, Candido-Ferreira I, Repapi E, Gavriouchkina D, Senanayake U, Ling ITC, Telenius J, Taylor S, Hughes J, Sauka-Spengler T | 2019 | Reconstruction of the Global Neural Crest Gene Regulatory Network In Vivo | https://www.ncbi.nlm.nih.gov/geo/query/acc.cgi?acc=GSE131688 | NCBI Gene Expression Omnibus, GSE131688 |

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
