## [Decision Letter]

**Decision letter after peer review:**

[Editors’ note: the authors submitted for reconsideration following the decision after peer review. What follows is the decision letter after the first round of review.]

Thank you for submitting the paper "Segregation of neural crest specific lineage trajectories from a heterogeneous neural plate border territory only emerges at neurulation" for consideration by *eLife*. Your article has been reviewed by 3 peer reviewers, and the evaluation has been overseen by a Reviewing Editor and a Senior Editor. The reviewers have opted to remain anonymous.

We are sorry to say that, after consideration of the reviewers' feedback and following further consultation with the reviewers, we have decided that this work will not be considered for publication by *eLife*. While this manuscript has not reached the benchmark of a Research Article, on noting the potential value of the dataset for future study of the development of the neuroectoderm, the neural crest cells and the ectodermal placode, we are keen to provide the opportunity for publishing this paper as Resource Article. Resubmission of a new manuscript would require addressing the critiques raised by the reviewers in this round of review, which are paraphrased below and outlined in the three reviewer reports.

The reviewers recognised that this study has collated a potentially useful dataset of the stage-resolved single-cell transcriptome of chick epiblast/ectoderm cells that may inform the development of the epiblast to neural folds and the molecular characteristics of the neural plate border cell population from which neuroepithelium, ectodermal neural placode and neural crest cells emerge. The findings reiterate the concept of lateral neural plate border harbours a heterogenous population of ectoderm cells and members of this population are precursors of the neural crest cell lineage during neurulation. The new knowledge gleaned from the transcriptome data is that the neural plate border cells are specified (and acquire the neural border signature, which includes novel marker genes identified in this study) progressively/dynamically during neurulation, but no insight is provided on how the neural border cells establish multipotency (e.g., re-gaining pluripotency factor activity). The data further showed that the segregation of the neural crest cell lineage and the placodal ectoderm lineage from the neural plate borders cells may occur at early neurulation, which is later than previously anticipated. Mining the transcriptome data of the emerging cell lineages has inferred the putative molecular drivers of the trajectory of neural plate border cells to neural crest cell lineage and placodal ectoderm lineage. There is no follow-up verification of the functional attributes of these drivers or the molecular network in the development of these two lineages. The merit of this study would be enhanced if functional analysis has been undertaken to identify which are the key drivers and their functional role in the segregation of the two divergent neural border cell lineages and the early steps of differentiation of the neural crest cells.

Re-submission as a resource report will require provision of the missing detail of the experimental design of the single-cell transcriptome analysis and the data analytics pipeline, improving the style of presentation to allow a better comprehension of the important novel findings of data mining. addressing the utility and limitation of the transcriptome dataset to gains knowledge that enables the formulation of testable hypothesis on the molecular activity driving neural plate development. emergence of the neural plate border cell lineages.

*Reviewer #1:*

The work by Williams et al. represents a significant effort to understand the lineages that emerge from the epiblast, which is the layer of cells that will give rise to the chick embryo, during the period from gastrulation through neurulation. Previously published work has addressed the question of when and where cells in the epiblast at the neural plate border become specified, but the authors strive to refine such data further and more precisely by using single-cell RNA-sequencing to characterize the signatures of these lineages in vivo. The authors focus on identifying cells that generate the placodes, neural crest, and neural tissues. Studying these populations of cells is well justified because of their relevance to human disease phenotypes particularly in the craniofacial complex, heart, gut, and malignancies more broadly. Another important contribution of the work is the descriptions of novel markers, which may ultimately result in identifying new genes associated with human disease.

A strength is that the manuscript is concise, clear, and well-written. Additionally, the figures are nicely organized and easy to follow. The images of embryonic gene expression (both known markers and validation of novel genes) are excellent in terms of tissue quality and confocal microscopy, and they provide robust support for the conclusions. The strategies employed for single-cell RNA-seq analysis represent a technical advance, and the results generated by this study will be of broad interest to the community. The use Hybridization Chain Reaction (HCR) with multiplexed probes to validate the RNA-seq dataset adds significant depth to the analysis.

A weakness is that the study is primarily descriptive with no experimental test of a hypothesis. For example, the work could benefit from some loss-of-function analyses for any of the novel genes identified. On this point, the authors have an excellent track record using high-throughput screening with chick embryos and morpholinos either in terms of characterizing gene regulatory networks (GRNs) or assaying for developmental phenotypes related to the neural crest. Moreover, work from the Bronner and Sauka-Spengler labs has previously contributed much in the way of our understanding of the GRNs that direct the delineation of the neural and non-neural lineages in the ectoderm. A discussion of how the current results fit into the larger framework of GRNs (both for known and novel genes) would provide a more complete context for the work. A schematic figure that maps these genes onto a GRN could be quite informative and clarifying.

Nonetheless, the results are significant in that they advance our understanding of the spatial and temporal expression of genes that are associated with cell specification and lineage restriction during the embryonic time course from gastrulation to neurulation. The conclusions are further supported by a modified RNA velocity analysis that reconstructs the temporal sequence of transcriptional steps and resolves gene-specific transcriptional dynamics over time. Importantly, based on this analysis, the authors find that cells from the emerging neural plate border are more heterogeneous than previously believed and they do not see the emergence of neural crest cells as a distinct lineage until later than what has been suggested by other studies.

Overall, the authors could do more in both the Introduction and the Discussion to emphasize exactly how their current study advances the field beyond simply describing the timing of specification and lineage restriction. This would be especially helpful since the authors cite earlier work from the Bronner lab (Roellig et al. 2017), which demonstrates that significant co-expression of neural crest, placode, and CNS markers persists in the neural plate border from gastrulation through neurulation. In fact, this earlier work has a similar conclusion that cells at the neural plate border contribute to multiple ectodermal lineages throughout neurulation and even after neural tube closure.

The sample sizes of embryos and numbers of cells collected are unclear. For example, the authors state that they isolate 2398 cells from 8 embryos (i.e., epiblasts?) at stage HH4. Does this total number consist of approximately 300 cells per embryo? The authors should explain why this number is sufficient as a representative and unbiased sample size in light of some reports that estimate up to 50,000 epithelial cells in the epiblast at this stage (see for example: Serrano Nájera G, Weijer CJ. Cellular processes driving gastrulation in the avian embryo. Mech Dev. 2020;163:103624). In the Methods section they state that "approximately 10,000 cells/stage were collected" and so it is hard to make sense of the sample sizes, etc. The authors should include the total number of embryos (n) for each RNA-seq analysis performed along with the number of cells (e.g., p. 15, lines 468-472).

Moreover, how can the authors claim with certainty that they are "profiling the majority of neural plate border cells" (p. 2, line 62)? Could there be other cells in the neural plate border that do not express Pax7 and Tfap2A? For each experiment, the authors should try to estimate how many cells (or what percentage) are not included in their analyses and discuss what the implications are for the interpretation of their results. A supplemental data table with total number of cells collected for each embryo at each stage in each experiment, would be helpful.

Given that HCR is also a quantitative technique and given that the authors use words like "stronger" or "decreased" to describe spatial and temporal changes in gene expression, the authors should try to measure levels (i.e., fold-differences) of expression among these markers and novel genes either relative to one another and/or over time.

Although the authors generate an intriguing list of genes that are co-expressed at the right time and place, they should refrain from equating "expression" with "function" (e.g., p. 13, line 371) given that the authors have not tested a role for any of the new genes that they have identified in their dataset. Also, the authors spend much of the Discussion describing previous experimental work for some of the genes included in their dataset. Here too, the authors should temper and contextualize their conclusions taking into account the fact that the presence of a specific transcript does not always correspond to actual gene function and protein activity, especially since there are numerous examples of post-transcriptional (and post-translational) regulation for many of these genes.

*Reviewer #2:*

This study performs single-cell transcriptome analysis of chicken embryos from gastrulation to neurulation stages (HH4-HH7) with the goal of understanding the emergence and transition of the neural plate border domain to placodal and neural crest lineages. The main impact of the study is the first detailed single-cell analysis of early chick embryos, which should complement similar studies in human, mouse, and other vertebrates. Major cell types are described at each stage, and this is coupled with in situ hybridization for a set of known and novel genes, as well as RNA velocity to deduce cell trajectories (based on the ratio of old cytoplasmic to new nuclear transcripts). Based on the absence of a distinct Pax7/Tfap2a cluster during gastrulation stages, the authors make an argument that the neural plate border and placode/neural crest does not emerge until later stages (i.e. neurulation), later than previously reported. However, the study largely falls short of making significant new insights into the timing and lineage trajectories of placodes and neural crest. The manuscript is very dense with descriptive data of gene expression in numerous clusters, which makes it difficult to extract big picture messages about lineage emergence. There are also numerous typos and figure errors that make the reading difficult. The RNA velocity analysis is not particularly convincing as to lineage emergence, and there are numerous overstatements about the ability of such single-cell genomics approaches to define in vivo lineages in the absence of experimental confirmation. While some new genes with neural plate border expression are presented, others are already well known or are shown to have very broad expression, thus limiting their utility in defining domains. There is a feeling that this work is quite preliminary in terms of its analysis of neural plate border lineages, though it is clearly an important resource for the chick embryology community.

1. Several statements are made as to lineage analysis, e.g. with relation to Pax7+ cells. Single-cell analysis can only make predictions about lineages, which need to be experimentally confirmed. These statements should therefore be removed or more carefully explained. For example, lines 51-52 (and also lines 276-277 and line 407) : "We further demonstrate that Pax7+ cells are not restricted to a neural crest fate and are capable of giving rise to all derivatives of the neural plate border."

2. In the Intro, it stated that the study identifies numerous novel factors, but the in situs presented show a combination of novel factors and many known factors (e.g. Dlx6, Msx1, Gbx2), with several of the "novel" genes (e.g. Ing5) being much broader than just prospective NPB territory. Similarly in Figure 4, many of the in situ markers are quite broad and therefore it is hard to determine their utility in defining specific domains (e.g. Gbx2 and Znf703). It is also an overstatement to say the study has identified key drivers of NPB trajectories in the absence of functional assays.

3. The known roles of Irf6 and Grhl3 in driving periderm fate should be better integrated in the single-cell analysis. The in situs confirm that these are expressed quite broadly and mainly in the overlying periderm.

4. The scVelo in Figure 5 is not particularly informative. From the various arrows and UMAPs shown, its hard to distill a clear theme of cell transitions. For example in panel F, how does the data specifically inform bifurcation of placodal and neural crest ectoderm?

5. What is "neural-neural crest"?

6. There appears to be an error in Figure 6A preventing the top left of the plot from being visualized. An image from panel D seems to be pasted by mistake. In general, there are many typos/errors in the manuscript that further make it difficult to follow.

7. It is unclear what the colors mean in Figure 6E,F. Figure S5I – the legend is similarly insufficient to understand the panels, e.g. no description of what colors means.

8. What is the evidence that mitochondrial genes at HH4 indicative of highly active migrating cells?

9. HH6-Cl2 is stated to be anterior lateral plate mesoderm, but Pitx2 and Alx1 are also markers of frontonasal neural crest mesenchyme. How do the authors distinguish between these possibilities?

10. In Figure S1B, ElalL1 and Tgfi1 are used as epiblast stem cell markers but appear to be very broadly expressed. Also in Figure S1F, the genes used to define the Node are not very specific.

11. The in situ in Figure S4D is not described in legend or annotated in actual panel.

*Reviewer #3:*

In this study, Williams and colleagues use single cell transcriptomics to describe the cell fates and lineages in the dorsal ectoderm of chick embryos taken from gastrula to early neurula stage. Specifically, they provide an atlas of whole embryo epiblast at stages 4,HH, 5HH, 6HH and 7H. They further subcluster the ectoderm cells and use cVelo algorithm (based on velocity of RNA splicing) to infer trajectories and lineages in this dataset.

Description of each stage and each cluster of cells is carefully done and retrieves the canonical markers for each territory. The neural plate border is found expressing pax7 early on, as previously described in chick embryos (Basch et al., 2006) but a specific clustering of these cells is not found, suggesting that the identity of the neural plate border is rather the overlap of gene signatures from adjacent tissues and that the neural crest and the placode signatures arise at the early neurula stage, in the elevating neural folds of the embryo.

From previous lineage tracing studies done at the gastrula stage, it was shown that the border between the prospective neural plate and the future non-neural ectoderm gives rise to four main cell fates: dorsal neural tube cells, neural crest cells, posterior placodal cells and non-neural ectoderm cells (e.g. Steventon et al., 2009). Genetically, this territory has previously been defined by the overlap between gene expressions defining the non-neural ectoderm and the neural ectoderm (e.g. De Crozé et al., 2011, Grooves and Labonne 2014)

Previous transcriptomic studies have pointed out the high similarity of the neural plate border territory compared to adjacent ectoderm regions, the failure of "classical" biostatistics tools to evidence a specific signature and the need for tailored strategies to do so (Plouhinec et al., 2017). Using those, the neural plate border could be defined from gastrula stage by its expression of pax3/pax7 ortholog and a couple of other genes, then by a more extended signature at neurulation stage.

In conclusion, although this study explores the same question with the latest tools of single cell transcriptomics, it is mostly descriptive and brings little novel insight into the biology of neural and neural plate border induction. However, it highlights a series of additional genes that could be of interest for further functional study.

In order to increase the impact of this study, functional analysis of some key genes should be added. Previous works from the same authors have elegantly shown how various regulators influence cell fates in the neural crest gene regulatory network.

[Editors’ note: further revisions were suggested prior to acceptance, as described below.]

Thank you for resubmitting your work entitled "Segregation of neural crest specific lineage trajectories from a heterogeneous neural plate border territory only emerges at neurulation" for further consideration by *eLife*. Your revised article has been evaluated by Kathryn Cheah (Senior Editor) and a Reviewing Editor.

The feedback of the reviewers indicate that the revision addresses most of original concerns and now fits better as a Resources paper. While the manuscript has been improved, there are some remaining issues that need to be addressed, as outlined below:

Consider modifying the title and summary to broaden the coverage of subject beyond the neural plate border cell population

Take into consideration of that the inference of molecular trajectory of cell population and the transcriptional network/s driving lineage development are hypothetical in the context of the origin of the neural crest cells at this juncture and tune the Discussion accordingly.

Address the critique of the data presented in Figures 5 and 6:

The computational results on the developmental trajectory (Figure 5) purportedly showed the connectivity of cell types at four developmental stages separately. As these are snapshots of the "transcriptome proximity" between cell types (whole epiblast/extracted ectoderm sub-clusters) at a single timepoint, it is not biological meaningful for inferring a developmental trajectory of cell population in time. These results (and the accompanying description of the data in Results) may be provided as supplementary information.

Enhance the visibility of the findings of scRNA velocity (Figure 6): The RNA velocity measurement across the four stages may be more appropriate for inferring the developmental trajectory of cell types across HH6-HH7. For more clarity, it may be helpful to highlight the predominant velocity vectors between the cell types, as some individual vectors appeared to be outliers (i.e. deviating from the global vectorial direction).

Please provide information on the development of the web portal and its state of accessibility. If possible, provide user feedback on a trial run to evaluate the utility and effectiveness of the web interface, which may be presented as supplementary information of revised manuscript.

---

## [Author Response]

[Editors’ note: the authors resubmitted a revised version of the paper for consideration. What follows is the authors’ response to the first round of review.]

[…] There is no follow-up verification of the functional attributes of these drivers or the molecular network in the development of these two lineages. The merit of this study would be enhanced if functional analysis has been undertaken to identify which are the key drivers and their functional role in the segregation of the two divergent neural border cell lineages and the early steps of differentiation of the neural crest cells.

Thank you for the useful comments. We agree that functional experiments would be very valuable and we have initiated some of these, but we also feel that given the myriad of new genes, finding the essential drivers will not be an easy task and would likely take an enormous amount of time execute in a controlled and validated fashion. Given the pandemic, we feel that these experiments would not be feasible in a reasonable amount of time. Moreover, the present manuscript is already overly long. We have endeavoured to make it clearer and more concise in the revised version. Therefore, we agree with the idea presented by the Senior Editor that the present dataset would make a very valuable Resource paper without adding functional data.

Re-submission as a resource report will require provision of the missing detail of the experimental design of the single-cell transcriptome analysis and the data analytics pipeline, improving the style of presentation to allow a better comprehension of the important novel findings of data mining. addressing the utility and limitation of the transcriptome dataset to gains knowledge that enables the formulation of testable hypothesis on the molecular activity driving neural plate development. emergence of the neural plate border cell lineages.

We thank the editor and reviewers for giving us the option to resubmit the paper as a Resource. Accordingly, we have amended the manuscript to address the very useful comments of the reviewers as described below. We are also in the process of uploading all our data sets to a web-based user interface such that readers can explore the data in its entirety.

Reviewer #1:[…] A discussion of how the current results fit into the larger framework of GRNs (both for known and novel genes) would provide a more complete context for the work. A schematic figure that maps these genes onto a GRN could be quite informative and clarifying.

We thank the reviewer for this insightful comment. To address this, we have added discussion of how some of the genes identified here fit into the broader neural crest gene regulatory network using previously published data (Williams et al., 2019) (lines 384-386 and 392-394). While the idea of putting some of the novel genes described here into a GRN is very attractive, it is premature in the absence of functional data since it is not possible to construct a GRN on expression data alone; this would require epistatic/regulatory data which is outside the scope of the current work.

Overall, the authors could do more in both the Introduction and the Discussion to emphasize exactly how their current study advances the field beyond simply describing the timing of specification and lineage restriction. This would be especially helpful since the authors cite earlier work from the Bronner lab (Roellig et al. 2017), which demonstrates that significant co-expression of neural crest, placode, and CNS markers persists in the neural plate border from gastrulation through neurulation. In fact, this earlier work has a similar conclusion that cells at the neural plate border contribute to multiple ectodermal lineages throughout neurulation and even after neural tube closure.

Thank you for noting this lack of information. To address this point, we have amended parts of the introduction and discussion as suggested. In particular we emphasize how our analysis expands on previous work (lines 38-41) and how our whole epiblast sc-RNA-seq data sets provide a contextual environment for understanding how the neural plate border emerges during early neurulation and diverges from the surrounding neural and non-neural ectoderm (line 381). We also further discuss how the genes identified here might fit into the broader neural crest GRN (lines 382-394) and we highlight the notion that neural plate border cells are not a unique population characterized by an exclusive gene expression signature, but rather that they share features with the surrounding tissues (lines 409-415).

The sample sizes of embryos and numbers of cells collected are unclear. For example, the authors state that they isolate 2398 cells from 8 embryos (i.e., epiblasts?) at stage HH4. Does this total number consist of approximately 300 cells per embryo? The authors should explain why this number is sufficient as a representative and unbiased sample size in light of some reports that estimate up to 50,000 epithelial cells in the epiblast at this stage (see for example: Serrano Nájera G, Weijer CJ. Cellular processes driving gastrulation in the avian embryo. Mech Dev. 2020;163:103624). In the Methods section they state that "approximately 10,000 cells/stage were collected" and so it is hard to make sense of the sample sizes, etc. The authors should include the total number of embryos (n) for each RNA-seq analysis performed along with the number of cells (e.g., p. 15, lines 468-472).Moreover, how can the authors claim with certainty that they are "profiling the majority of neural plate border cells" (p. 2, line 62)? Could there be other cells in the neural plate border that do not express Pax7 and Tfap2A? For each experiment, the authors should try to estimate how many cells (or what percentage) are not included in their analyses and discuss what the implications are for the interpretation of their results.

We apologize for this lack of clarity. We have now added the number of embryos and cells used in the analysis to the text (lines 71, 84, 97 and 102) and also provide a supplemental table detailing the number of embryos/cells used at each stage. Along with the CellRanger output.

It is likely that the epiblast contains up to 50,000 cells; however approximately 50% of cells may be lost during the dissociation/FACS process. Furthermore, the 10X Chromium platform has limited loading capacity (46.6ul), and as it is not advisable to spin down cells and resuspend in a smaller volume, the number cells loaded is restricted. Moreover, the 10X Chromium platform can only recover up to 10,000 cells, so overloading is not advantageous. The percentage of cells we recover is largely within the 10X guidelines of 45-60%, the exception was HH4 (30%) however this was performed with v2 10X chemistry.

Given that we resolve numerous clusters that clearly represent each germ layer and sub-divisions thereof, we believe these are sufficient numbers of cells to construct an unbiased single-cell atlas at each stage. Moreover, we record high quality gene expression metrics indicating the cells were of good quality (see supplemental table). We have now clarified these points in the text and apologize that this was not clearer in the first version of the manuscript.

A supplemental data table with total number of cells collected for each embryo at each stage in each experiment, would be helpful.

Thank you for this excellent suggestion. This is now provided in the supplemental section.

Given that HCR is also a quantitative technique and given that the authors use words like "stronger" or "decreased" to describe spatial and temporal changes in gene expression, the authors should try to measure levels (i.e., fold-differences) of expression among these markers and novel genes either relative to one another and/or over time.

We appreciate the reviewers point here. While HCR can be quantitative if performed at nonsaturating levels, embryos must be processed concomitantly and under exactly the same conditions to make accurate comparisons. As we did not prepare the whole mount embryos in this way, it would be no trivial undertaking to repeat all the HCR’s, imaging and quantification. Furthermore, in the broader context of this work, the HCR analysis was done to provide validation of novel genes identified in the sc-RNA-seq data sets, and to uncover putative candidates for further investigation. However, we recognize the need for quantification here, as such we now provide the fold change values where applicable from the sc-RNA-seq data for genes shown in HCR. This information has been added on the relevant feature plots in figures 3 and 4. We have also altered our wording accordingly such that our description refers to qualitative rather than quantitative changes in gene expression levels (lines 181, 183, 198, 200, 208, 225).

Although the authors generate an intriguing list of genes that are co-expressed at the right time and place, they should refrain from equating "expression" with "function" (e.g., p. 13, line 371) given that the authors have not tested a role for any of the new genes that they have identified in their dataset. Also, the authors spend much of the Discussion describing previous experimental work for some of the genes included in their dataset. Here too, the authors should temper and contextualize their conclusions taking into account the fact that the presence of a specific transcript does not always correspond to actual gene function and protein activity, especially since there are numerous examples of post-transcriptional (and post-translational) regulation for many of these genes.

We thank the reviewer for raising this concern. We have now modified the text (line 58, 382, 387, 389-391, 397) to be more circumspect. We have also added references here to support our discussion of previous work (lines 207, 225, 389-391, 395-397).

Reviewer #2:[…]1. Several statements are made as to lineage analysis, e.g. with relation to Pax7+ cells. Single-cell analysis can only make predictions about lineages, which need to be experimentally confirmed. These statements should therefore be removed or more carefully explained. For example, lines 51-52 (and also lines 276-277 and line 407) : "We further demonstrate that Pax7+ cells are not restricted to a neural crest fate and are capable of giving rise to all derivatives of the neural plate border."

We thank the reviewer for this comment and agree that it is important to distinguish between gene expression and lineage relationships. Accordingly, we have adjusted the text to dampen down these claims (lines 56-57, 280-281, 284, and 435). Indeed, this was a most interesting finding which we intend to follow up experimentally and include in future publications.

2. In the Intro, it stated that the study identifies numerous novel factors, but the in situs presented show a combination of novel factors and many known factors (e.g. Dlx6, Msx1, Gbx2), with several of the "novel" genes (e.g. Ing5) being much broader than just prospective NPB territory. Similarly in Figure 4, many of the in situ markers are quite broad and therefore it is hard to determine their utility in defining specific domains (e.g. Gbx2 and Znf703). It is also an overstatement to say the study has identified key drivers of NPB trajectories in the absence of functional assays.

The reviewer raises an interesting point. Indeed, we believe it is a combination of transcription and signaling factors (both novel and known), that are involved in refinement of the neural plate border. For our in situ analysis, we combined probes to known neural plate border genes *(Tfap2A, Pax7, Msx1*) genes together with novel genes to observe co-expression in this region. We have now added this information in the text (line 170-171). We appreciate that the expression pattern of some of the genes shown here have previously been described, but not in the context of neural plate border development. We have attenuated the text accordingly and added citations (line 207 and 225).

Indeed, many of the genes explored in this study are expressed more broadly than the neural plate border, in keeping with the notion that the neural plate border is not defined by an exclusive set of genes. Rather, it is the combination of genes co-expressed across this region that are differentially shared with surrounding tissues. We have reinforced this point in the text line (lines 409-415). We have also toned down the text regarding key drivers of specific trajectories (lines 58, 340-341).

3. The known roles of Irf6 and Grhl3 in driving periderm fate should be better integrated in the single-cell analysis. The in situs confirm that these are expressed quite broadly and mainly in the overlying periderm.

We now mention the roles of *Irf6* and *Grhl3* in periderm formation and indeed Van der Woude syndrome. We thank the reviewer for bringing this information to our attention (lines 388-391).

4. The scVelo in Figure 5 is not particularly informative. From the various arrows and UMAPs shown, its hard to distill a clear theme of cell transitions. For example in panel F, how does the data specifically inform bifurcation of placodal and neural crest ectoderm?

Thank you for pointing out that this was not sufficiently clear. We have attempted to improve the visualisation of scVelo analysis by showing alternative plots in Figures 5 and 6. In these the arrows represent an average velocity across given clusters such that fewer but bolder arrows are shown. We have also removed the shading around the Pax7+ zone in Figure 5F as *Pax7* expression is demonstrated in the feature plots shown alongside (Figure 5G). We hope this makes this point clearer. We have also further clarified this in the text (lines 279-281).

5. What is "neural-neural crest"?

We apologise for not making this description clear. The split in neural crest cells between neural potential and mesenchymal potential has previously been described in this way (Williams et al. 2019) and we now clarify this in the text (line 293-296).

6. There appears to be an error in Figure 6A preventing the top left of the plot from being visualized. An image from panel D seems to be pasted by mistake. In general, there are many typos/errors in the manuscript that further make it difficult to follow.

We apologise for this problem which we failed to notice in the previous submission. Similarly, we have endeavoured to identify and correct other typos and figure errors. We thank the reviewer for pointing out this problem.

7. It is unclear what the colors mean in Figure 6E,F. Figure S5I – the legend is similarly insufficient to understand the panels, e.g. no description of what colors means.

Thank you for pointing out this lack of clarity here. We now describe this better and define the meaning of the colors in the figure legends. We have also added scale bars on the figures and describe the color correlation in the figure legends.

8. What is the evidence that mitochondrial genes at HH4 indicative of highly active migrating cells?

The reviewer raises a good point. There is no evidence that mitochondrial genes indicate active migration, but since we also detected cell migration genes in the same cluster, we thought the high levels of mitochondrial activity may represent highly active migrating cells. However, we recognize this is an over-interpretation and have removed this statement (line 445-446).

9. HH6-Cl2 is stated to be anterior lateral plate mesoderm, but Pitx2 and Alx1 are also markers of frontonasal neural crest mesenchyme. How do the authors distinguish between these possibilities?

Interesting point. We now clarify that HH6-Cl2 is also enriched for other lateral plate mesoderm (LPM) genes, including; *ShisA2, Sfrp1, OlfmL3, Six1* and *Twist1*. While some of these genes are indeed expressed in other tissues, collectively their shared expression is most dominant in the LPM. We have now added these examples to the text in the supplementary Results section (line 450). If the reviewers and editor so desire, we could add HCR of Pitx2 and Alx1 to clarify their location at HH6.

10. In Figure S1B, ElalL1 and Tgfi1 are used as epiblast stem cell markers but appear to be very broadly expressed. Also in Figure S1F, the genes used to define the Node are not very specific.

We agree that *Elav1* and *Tgif* are broadly expressed, but they are enriched 0.479 and 0.472 average fold change, respectively, in HH4-Cl2. We now include this information on the feature plots in Figure. S1B. We had difficultly annotating HH5-Cl7 (Figure S1F) and we agree the genes shown on the heatmap are not indicative of the node. We originally assigned this cluster as node based on less enriched genes not shown here. We have now relabeled this cluster as ‘Miscellaneous’ since the top enriched genes represent a myriad of cellular processes.

11. The in situ in Figure S4D is not described in legend or annotated in actual panel.

We thank the reviewer for noting this over sight. We have now annotated the in situ image and describe it in the legend.

Reviewer #3:[…]In order to increase the impact of this study, functional analysis of some key genes should be added. Previous works from the same authors have elegantly shown how various regulators influence cell fates in the neural crest gene regulatory network.

We thank the reviewer for their comments and appreciate the fact that it would be ideal to include functional experiments. That said, the manuscript is already overly long. More importantly, it would take us +1 year to test the new transcription factors described here and assemble them into a meaningful gene regulatory network. Thus, we would prefer to publish the paper as a Resource (as suggested) and follow-up with detailed functional analysis at a later time.

References

Williams, R. M., Candido-Ferreira, I., Repapi, E., Gavriouchkina, D., Senanayake, U., Ling, I. T. C., Telenius, J., Taylor, S., Hughes, J. and Sauka-Spengler, T. 2019. Reconstruction of the Global Neural Crest Gene Regulatory Network in vivo. *Dev Cell,* 51**,** 255-276 e7.

[Editors’ note: what follows is the authors’ response to the second round of review.]

The feedback of the reviewers indicate that the revision addresses most of original concerns and now fits better as a Resources paper. While the manuscript has been improved, there are some remaining issues that need to be addressed, as outlined below:Consider modifying the title and summary to broaden the coverage of subject beyond the neural plate border cell population

We have now changed the title and edited the text in the abstract and introduction to describe this study more broadly.

Take into consideration of that the inference of molecular trajectory of cell population and the transcriptional network/s driving lineage development are hypothetical in the context of the origin of the neural crest cells at this juncture and tune the Discussion accordingly.

We have adjusted the text accordingly to reflect the hypothetical nature of lineage trajectory analysis.

Address the critique of the data presented in Figures 5 and 6:The computational results on the developmental trajectory (Figure 5) purportedly showed the connectivity of cell types at four developmental stages separately. As these are snapshots of the "transcriptome proximity" between cell types (whole epiblast/extracted ectoderm sub-clusters) at a single timepoint, it is not biological meaningful for inferring a developmental trajectory of cell population in time. These results (and the accompanying description of the data in Results) may be provided as supplementary information.

We have now moved this data and accompanying text to the supplemental section, this now makes supplemental figure 6. The main manuscript is now 5 figures and we have adjusted the figure citations throughout.

Enhance the visibility of the findings of scRNA velocity (Figure 6): The RNA velocity measurement across the four stages may be more appropriate for inferring the developmental trajectory of cell types across HH6-HH7.

We have included scVelo data from stages HH5/6/7/8. We did not find that including HH4 and excluding HH8 was useful.

For more clarity, it may be helpful to highlight the predominant velocity vectors between the cell types, as some individual vectors appeared to be outliers (i.e. deviating from the global vectorial direction).

We appreciate the reviewers remark however, having already changed the velocity plots here (now figure 5) to the big, bold, averaged arrows to be clearer, we feel adding any more annotation will only further complicate the results as there is already lots of colour and arrows we don’t feel more annotation would help.

Please provide information on the development of the web portal and its state of accessibility. If possible, provide user feedback on a trial run to evaluate the utility and effectiveness of the web interface, which may be presented as supplementary information of revised manuscript.

All the data from this study is now available on the Single Cell Portal. This has been trialed by colleagues and corrected accordingly. This will be made publicly available once the paper is formally accepted.